# STEP-WISE TRIPLE-CONSISTENT DIFFUSION SAMPLING FOR INVERSE PROBLEMS

## ABSTRACT

Diffusion models (DMs) are a class of generative models that allow sampling from a distribution learned over a training set. When applied to solving inverse imaging problems (IPs), the reverse sampling steps of DMs are typically modified to approximately sample from a measurement-conditioned distribution in the image space. However, these modifications may be unsuitable for certain settings (such as in the presence of measurement noise) and non-linear tasks, as they often struggle to correct errors from earlier sampling steps and generally require a large number of optimization and/or sampling steps. To address these challenges, we state three conditions for achieving measurement-consistent diffusion trajectories. Building on these conditions, we propose a new optimization-based sampling method that not only enforces the standard data manifold measurement consistency and forward diffusion consistency, as seen in previous studies, but also incorporates step-wise and network-regularized backward diffusion consistency that maintains a diffusion trajectory by optimizing over the input of the pre-trained model at every sampling step. By enforcing these conditions, either implicitly or explicitly, our sampler requires significantly fewer reverse steps. Therefore, we refer to our accelerated method as **S**tep-w**i**se **T**riple-**Co**nsistent Sa**m**pling (**SITCOM**). Compared to existing state-of-the-art baseline methods, under different levels of measurement noise, our extensive experiments across five linear and three non-linear image restoration tasks demonstrate that SITCOM achieves competitive or superior results in terms of standard image similarity metrics while requiring a reduced run-time across all considered tasks.

## 1 INTRODUCTION

Inverse problems (IPs) arise in a wide range of science and engineering applications, including computer vision (Li et al., 2024), signal processing (Byrne, 2003), medical imaging (Alkhouri et al., 2024), remote sensing (Levis et al., 2022), and geophysics (BniLam and Al-Khoury, 2020). In these applications, the primary goal is to recover an unknown image or signal $\mathbf{x} \in \mathbb{R}^n$ from measurements or degraded image $\mathbf{y} \in \mathbb{R}^m$, which are often corrupted by noise. Mathematically, the unknown signal and the measurements are related as

$$\mathbf{y} = \mathcal{A}(\mathbf{x}) + \mathbf{n}, \tag{1}$$

where $\mathcal{A}(\cdot) : \mathbb{R}^n \rightarrow \mathbb{R}^m$ (with $m \leq n$) represents the linear or non-linear forward operator that models the measurement process, and $\mathbf{n} \in \mathbb{R}^m$ denotes the noise in the measurement domain, e.g., assumed sampled from a Gaussian distribution $\mathcal{N}(\mathbf{0}, \sigma_{\mathbf{y}}^2 \mathbf{I})$, where $\sigma_{\mathbf{y}} > 0$ denotes the noise level. Exactly solving these inverse problems is challenging due to their ill-posedness in many settings, requiring advanced techniques to achieve accurate solutions.

Deep learning techniques have recently been utilized as a prior to aid in solving these problems (Ravishankar et al., 2019; Lempitsky et al., 2018). One framework that has shown significant potential is the use of generative models, particularly diffusion models (DMs) (Ho et al., 2020). Given a training dataset, DMs are trained to learn the underlying distribution $p(\mathbf{x})$. During inference, DMs enable sampling from this learned distribution through an iterative procedure (Song et al., 2021b). When employed to solving inverse problems, DM-based IP solvers often modify the reverse sampling steps to allow sampling from the measurements-conditioned distribution $p(\mathbf{x}|\mathbf{y})$ (Chung et al.,

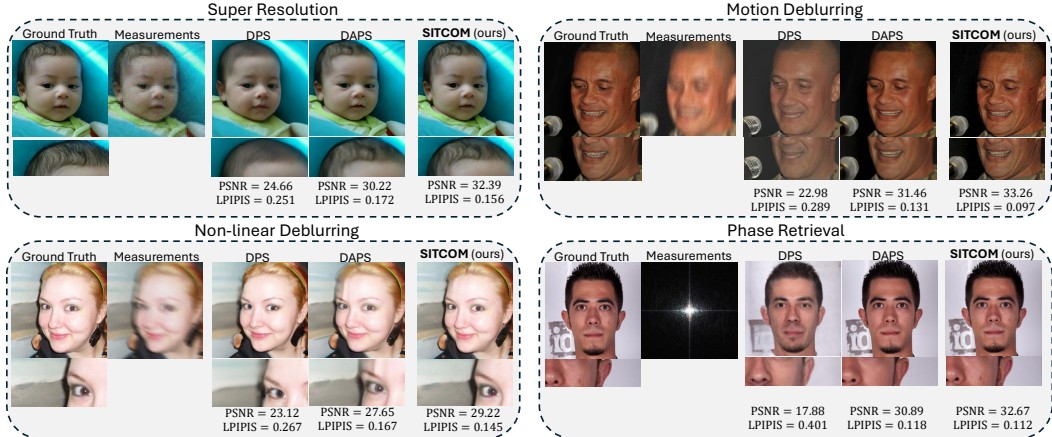

Figure 1: **Qualitative results on the FFHQ dataset** on two linear tasks (*top*) and two non-linear tasks (*bottom*) under measurement noise of $\sigma_{\mathbf{y}} = 0.05$. The PSNR and LPIPS values are given below each restored image. Zoomed-in regions show how SITCOM captures greater image details when compared to two general (non)linear DM-based methods (DPS (Chung et al., 2023b) and DAPS (Zhang et al., 2024)).

2023b; 2022). These modifications typically rely on approximations that may not be suitable for all tasks and settings, and in addition to generally requiring many sampling iterations, often suffer from errors accumulated during early diffusion sampling steps (Zhang et al., 2024). In most DM-based IP solvers, these approximations are designed to enforce standard measurement consistency on the estimated image (or posterior mean) at every reverse sampling iteration, as in (Chung et al., 2023b), and may also include resampling using the forward diffusion process (which we refer to as forward diffusion consistency), such as in (Lugmayr et al., 2022; Song et al., 2023a).

A key bottleneck in DMs is their computational speed, as they are slower than other generative models due to the large number of sampling steps. Although various methods have been proposed to reduce sampling frequency (e.g., (Song et al., 2023b)), these improvements have yet to be fully realized for DMs applied to IPs. Most existing methods still require dense sampling, which continues to pose speed challenges.

**Contributions:** In this paper, we: (*i*) identify key issues in accelerating DMs for IPs, (*ii*) propose three conditions that could fully leverage the information from the measurements and the implicit bias of the pre-trained DM to effectively address these issues, and (*iii*) present a new optimization-based sampler that satisfies these conditions. We refer to our accelerated sampling method as **S**tep-**w**ise **T**riple-**Co**nsistent Sa**m**pling (SITCOM). We evaluate our method on several image restoration tasks. Compared to leading baselines, our approach consistently achieves either state-of-the-art or highly competitive quantitative results, while also reducing the number of sampling steps and, consequently, the computational time. See Figure 1 for examples.

## 2 BACKGROUND: DIFFUSION MODELS & THEIR USAGE IN SOLVING IPS

Pre-trained Diffusion Models (DMs) generate images by applying a pre-defined iterative denoising process (Ho et al., 2020). In the Variance-Preserving Stochastic Differentiable Equations (SDEs) setting (Song et al., 2021b;a), DMs are formulated using the forward and reverse processes

$$d\mathbf{x}_t = -\frac{\beta_t}{2}\mathbf{x}_t dt + \sqrt{\beta_t}d\mathbf{w}, \quad d\mathbf{x}_t = -\beta_t\left[\frac{1}{2}\mathbf{x}_t + \nabla_{\mathbf{x}_t}\log p_t(\mathbf{x}_t)\right]dt + \sqrt{\beta_t}d\bar{\mathbf{w}}, \quad (2)$$

where $\beta : \{0, \dots, T\} \to (0, 1)$ is a pre-defined function that controls the amount of additive perturbations at time $t$, $\mathbf{w}$ (resp. $\bar{\mathbf{w}}$) is the forward (resp. reverse) Weiner process (Anderson, 1982), $p_t(\mathbf{x}_t)$ is the distribution of $\mathbf{x}_t$ at $t$, and $\nabla_{\mathbf{x}_t}\log p_t(\mathbf{x}_t)$ is the score function that is replaced by a neural network (typically a time-encoded U-Net (Ronneberger et al., 2015)) $\boldsymbol{s} : \mathbb{R}^n \times \{0, \dots, T\} \to \mathbb{R}^n$, parameterized by $\theta$. In practice, given the score function $\boldsymbol{s}_\theta$, the SDEs in Equation (2) can be discretized as in Equation (3) where $\boldsymbol{\eta}_t, \boldsymbol{\eta}_{t-1} \sim \mathcal{N}(\mathbf{0}, \mathbf{I})$.

$$\mathbf{x}_t = \sqrt{1-\beta_t}\mathbf{x}_{t-1} + \sqrt{\beta_t}\boldsymbol{\eta}_{t-1}, \quad \mathbf{x}_{t-1} = \frac{1}{\sqrt{1-\beta_t}}\left[\mathbf{x}_t + \beta_t\boldsymbol{s}_\theta(\mathbf{x}_t, t)\right] + \sqrt{\beta_t}\boldsymbol{\eta}_t. \quad (3)$$

When employed to solve inverse problems, the score function in Equation (2) is replaced by a conditional score function which, by Bayes' rule, is $\nabla_{\mathbf{x}_t} \log p_t(\mathbf{x}_t|\mathbf{y}) = \nabla_{\mathbf{x}_t} \log p_t(\mathbf{x}_t) + \nabla_{\mathbf{x}_t} \log p_t(\mathbf{y}|\mathbf{x}_t)$. Solving the SDE in Equation (2) with the conditional score is referred to as *posterior sampling* (Chung et al., 2023b). As there doesn't exist a closed-form expression for the term $\nabla_{\mathbf{x}_t} \log p_t(\mathbf{y}|\mathbf{x}_t)$ (which is termed as the measurements matching term in (Daras et al., 2024)), previous works have explored different approaches, which we will briefly discuss below. We refer the reader to the recent survey in (Daras et al., 2024) for an overview on DM-based methods for solving IPs.

A well-known method is Diffusion Posterior Sampling (DPS) (Chung et al., 2023b), which uses the approximation $p(\mathbf{y}|\mathbf{x}_t) \approx p(\mathbf{y}|\hat{\mathbf{x}}_0)$ where $\hat{\mathbf{x}}_0(\mathbf{x}_t)$ (or simply $\hat{\mathbf{x}}_0$) is the estimated image at time $t$ as a function of the pre-trained model and $\mathbf{x}_t$ (Tweedie's formula (Vincent, 2011)), given as

$$\hat{\mathbf{x}}_0(\mathbf{x}_t) = \frac{1}{\sqrt{\bar{\alpha}_t}}\Big[\mathbf{x}_t - \sqrt{1-\bar{\alpha}_t}\boldsymbol{\epsilon}_\theta(\mathbf{x}_t, t)\Big] =: f(\mathbf{x}_t; t, \boldsymbol{\epsilon}_\theta) \,, \tag{4}$$

where $\bar{\alpha}_t = \prod_{j=1}^{t} \alpha_j$ and $\alpha_t = 1 - \beta_t$. We call the function $f$, defined in Equation (4), as **'Tweedie-network denoiser'** (also termed as 'posterior mean predictor' in (Chen et al., 2024)). Here, $\boldsymbol{\epsilon}_\theta(\mathbf{x}_t, t) = -\sqrt{1-\bar{\alpha}_t}\boldsymbol{s}_\theta(\mathbf{x}_t, t)$ (Luo, 2022) outputs the noise in $\mathbf{x}_t$. Tweedie's formula is also adopted in other DM-based IP solvers such as (Rout et al., 2023; Chung et al., 2023c; Wang et al., 2022). The drawback of these methods is that they require a large number of sampling steps.

The work in ReSample (Song et al., 2023a), solves an optimization problem on the estimated posterior mean in the latent space to enforce a step-wise measurement consistency, requiring many sampling and optimization steps.

The work in (Mardani et al., 2023) introduced RED-Diff, a variational Bayesian method that fits a Gaussian distribution to the posterior distribution of the clean image given the measurements. This approach involves solving an optimization problem using stochastic gradient descent (SGD) to minimize a data-fitting term while maximizing the likelihood of the reconstructed image under the denoising diffusion prior (as a regularizer). However, the SGD process requires multiple iterations, each involving evaluations of the pre-trained DM on a different noisy image at some randomly selected time. While RED-diff reduces the run-time, their qualitative results are not competitive on several image restoration tasks.

Recently, Decoupling Consistency with Diffusion Purification (DCDP) (Li et al., 2024) proposed separating diffusion sampling steps from measurement consistency by using DMs as diffusion purifiers (Nie et al., 2022; Alkhouri et al., 2024), with the goal of reducing the run-time. However, for every task, DCDP requires tuning the number of forward diffusion steps for purification for each sampling step. Shortly after, Decoupled Annealing Posterior Sampling (DAPS) (Zhang et al., 2024) introduced another decoupled approach, incorporating gradient descent noise annealing via Langevin dynamics. DAPS, similar to DPS, also requires a large number of sampling and optimization steps. Under measurement noise, DCDP achieves SOTA run-time across various linear restoration tasks, while DAPS sets the SOTA in restoration quality. Both will serve as primary baselines in our experiments.

## 3 SITCOM: STEP-WISE TRIPLE-CONSISTENT SAMPLING

### 3.1 MOTIVATION: ADDRESSING THE CHALLENGES IN APPLYING DMS TO IPS

Most inverse problems are ill-conditioned and undersampled. DMs, when trained on a dataset that closely resembles the target image, can provide critical information to alleviate ill-conditioning and improve recovery. Despite various previous efforts, a key challenge remains: How to *efficiently* integrate DMs into the framework of inverse problems? We will now elaborate on this challenge in detail.

The standard reverse sampling procedure in DMs consists of applying the backward discrete steps in Equation (3) for $t \in \{T, T-1, \ldots, 1\}$, forming the standard diffusion trajectory for which $\mathbf{x}_0$ is the generated image[1]. To incorporate the measurement $\mathbf{y}$ into these steps, a common approach

---

[1]Diffusion trajectory refers to the path that leads to an in-distribution image, where the distribution is the one learned by the DM from the training set.

adopted in previous works that demonstrate superior performance (e.g., (Song et al., 2023a; Zhang et al., 2024; Li et al., 2024)) is to the $\hat{\mathbf{x}}_0$ computed via Equation (4) as follows:

$$\hat{\mathbf{x}}_0'(\mathbf{x}_t) = \arg\min_{\mathbf{x}} \|\mathcal{A}(\mathbf{x}) - \mathbf{y}\|^2 + \lambda\|\mathbf{x} - \hat{\mathbf{x}}_0(\mathbf{x}_t)\|^2 , \quad (5)$$

where $\lambda \in \mathbb{R}_+$ is a regularization parameter. The $\hat{\mathbf{x}}_0'(\mathbf{x}_t)$ obtained from Equation (5) is close to $\hat{\mathbf{x}}_0(\mathbf{x}_t)$ while also remaining consistent with the measurements. When using $\hat{\mathbf{x}}_0'(\mathbf{x}_t)$ to sample $\mathbf{x}_{t-1}$, the second formula in Equation (3) can be rewritten as in Equation (6), where the derivation is provided in Appendix A.

$$\mathbf{x}_{t-1} = \frac{\sqrt{\alpha_t}(1 - \bar{\alpha}_{t-1})}{1 - \bar{\alpha}_t}\mathbf{x}_t + \frac{\sqrt{\bar{\alpha}_{t-1}}\beta_t}{1 - \bar{\alpha}_t}\hat{\mathbf{x}}_0(\mathbf{x}_t) + \sqrt{\beta_t}\boldsymbol{\eta}_t . \quad (6)$$

By substituting the $\hat{\mathbf{x}}_0(\mathbf{x}_t)$ in Equation (6) with the measurement-consistent $\hat{\mathbf{x}}_0'(\mathbf{x}_t)$, the modified sampling formula becomes:

$$\mathbf{x}_{t-1} = \frac{\sqrt{\alpha_t}(1 - \bar{\alpha}_{t-1})}{1 - \bar{\alpha}_t}\mathbf{x}_t + \frac{\sqrt{\bar{\alpha}_{t-1}}\beta_t}{1 - \bar{\alpha}_t}\hat{\mathbf{x}}_0'(\mathbf{x}_t) + \sqrt{\beta_t}\boldsymbol{\eta}_t . \quad (7)$$

While this approach effectively ensures data consistency at each step, it inevitably causes $\hat{\mathbf{x}}_0'$ to deviate from the diffusion trajectory, leading to two major issues:

**(I1)** The image $\hat{\mathbf{x}}_0(\mathbf{x}_t)$, initially constructed through Tweedie's formula, usually appears quite natural (e.g., columns 3 to 5 of Figure 2 ); however, the modified version, $\hat{\mathbf{x}}_0'(\mathbf{x}_t)$, is likely to exhibit severe artifacts (e.g., columns 6 to 8 of Figure 2).

**(I2)** Since the DM network, $\boldsymbol{\epsilon}_\theta$, is trained via minimizing the objective function $\mathbb{E}_{\mathbf{x}_0,\boldsymbol{\epsilon}}\|\boldsymbol{\epsilon} - \boldsymbol{\epsilon}_\theta(\sqrt{\bar{\alpha}_t}\mathbf{x}_0 + \sqrt{1 - \bar{\alpha}_t}\boldsymbol{\epsilon}, t)\|^2$ (denoising score matching (Vincent, 2011)) on a finite dataset, it performs best on noisy images lying in the high-density regions of the *training distribution* $\mathcal{N}(\mathbf{x}_t; \sqrt{\bar{\alpha}_t}\mathbf{x}_0, (1 - \bar{\alpha}_t)\mathbf{I})$, $\mathbf{x}_0 \sim p(\mathbf{x}_0)$. We define an algorithm as **forward-consistent** if it likely applies $\boldsymbol{\epsilon}_\theta$ only to in-distribution inputs (i.e., those from the same distribution used for training). For example, if the forward diffusion used to train $\boldsymbol{\epsilon}_\theta$ adds Gaussian noise, the in-distribution input to $\boldsymbol{\epsilon}_\theta$ should ideally be sampled from a Gaussian with specific parameters. If Poisson noise is used in the forward process, inputs drawn from suitable Poisson distributions are more likely to fall within the well-trained region of the network. In summary, forward consistency requires that inputs to $\boldsymbol{\epsilon}_\theta$ during sampling align with the forward process. While the $\mathbf{x}_{t-1}$ generated from Equation (6) is forward-consistent by design, the one generated from the modified formula Equation (7) is not. Therefore, in the latter case, the DM network, $\boldsymbol{\epsilon}_\theta$, may be applied to many out-of-distribution inputs, leading to degraded performance.

We pause to verify our claimed Issue **(I1)** through a box-inpainting experiment. Columns 3 to 5 of Figure 2 show $\hat{\mathbf{x}}_0'(\mathbf{x}_t)$ at various $t$. The results clearly demonstrate successful enforcement of data consistency, as the region outside the box aligns with the original image. However, this enforcement compromises the natural appearance of the image, introducing significant artifacts in the reconstructed area inside the box. Details about the setting of the results in Figure 2 are given in Section C.

Issue **(I2)** was previously observed in (Lugmayr et al., 2022), which proposed a remedy known as '*resampling*'. In this approach, the sampling formula in Equation (7) is replaced by

$$\mathbf{x}_{t-1} = \sqrt{\bar{\alpha}_{t-1}}\hat{\mathbf{x}}_0 + \sqrt{1 - \bar{\alpha}_{t-1}}\boldsymbol{\eta}_t . \quad (8)$$

Provided $\hat{\mathbf{x}}_0$ is close to the ground truth $\mathbf{x}_0$, the $\mathbf{x}_{t-1}$ generated this way will stay in-distribution with high probability. For a more detailed explanation of the rationale behind this remedy, we refer the reader to (Lugmayr et al., 2022). This method has since been adopted by subsequent works, such as (Song et al., 2023a; Zhang et al., 2024), and we will also employ it to address **(I2)**.

## 3.2 NETWORK REGULARIZATION & BACKWARD DIFFUSION CONSISTENCY

Previous studies, such as (Song et al., 2023a; Zhang et al., 2024), mitigate issue **(I1)** by using a large number of sampling steps, which inevitably increases the computational burden. In contrast, this paper proposes employing a *network regularization* to resolve issue **(I1)**. This approach not only

Figure 2: Effects of enforcing backward-consistency in box-inpainting: Results of using Tweedie's formula without measurement consistency (columns 3 to 5), enforcing measurement-consistency via Equation (5) (columns 6 to 9), and enforcing both measurement-consistency and backward-consistency via Equation (12) (columns 10 to 12) at different time steps $t'$. Experimental details are given in Appendix C.

accelerates convergence but also enhances reconstruction quality. Let's first clarify the underlying intuition.

It is widely observed that the U-Net architecture or trained transformers exhibit an effective image bias (Ulyanov et al., 2018; Liang et al., 2024; Ghosh et al., 2024; Hatamizadeh et al., 2023). From columns 3 to 5 of Figure 2, we observe that without enforcing data consistency, the reconstructed $\hat{\mathbf{x}}_0$, derived directly from Tweedie-network denoiser $f(\mathbf{x}_t; t, \boldsymbol{\epsilon}_\theta)$ for each time $t$, exhibits natural textures. This indicates that the reconstruction using the combination of Tweedie's formula and the DM network has a natural regularizing effect on the image.

By definition, the output of $f(\mathbf{x}_t; t, \boldsymbol{\epsilon}_\theta)$ in Equation (4) represents the *denoised* version of $\mathbf{x}_t$ at time $t$ using the Tweedie's formula and the DM denoiser $\boldsymbol{\epsilon}_\theta$. Due to the implicit bias of $\boldsymbol{\epsilon}_\theta$, this denoised image tends to align with the clean image manifold, even if $\mathbf{x}_t$ does not correspond to a training image, as shown in columns 3 to 5 of Figure 2. We refer to this regularization effect of $f(\mathbf{x}_t; t, \boldsymbol{\epsilon}_\theta)$, which arises from network bias, as 'network regularization'.

By employing network regularization, we can address **(I1)** by ensuring that the data-consistent $\hat{\mathbf{x}}_0'$ is also network-consistent. We refer the latter condition as **Backward Consistency** and define it formally as follows.

**Definition 1** (Backward Consistency). *We say a reconstruction $\hat{\mathbf{x}}_0'$ is backward-consistent with posterior mean predictor $f(\,\cdot\,; t, \boldsymbol{\epsilon}_\theta)$ at time $t$ if it can be expressed as $\hat{\mathbf{x}}_0' = f(\mathbf{v}_t; t, \boldsymbol{\epsilon}_\theta)$ with some $\mathbf{v}_t$. In other words, backward consistency requires $\hat{\mathbf{x}}_0'$ to be an output of $f$ at time $t$.*

The use of network regularization to define step-wise backward consistency is inspired by the implicit bias of the Deep Image Prior (DIP) (Ulyanov et al., 2018). When $g_\phi$ represents a DIP with $\phi$ as its weights, it can regularize the reconstruction of inverse problems by solving the following optimization problem: $\hat{\phi}, \hat{\mathbf{x}} = \{\arg\min_{\phi, \mathbf{x}} \|\mathcal{A}(\mathbf{x}) - \mathbf{y}\|_2^2, \quad \text{subject to} \quad \mathbf{x} = g_\phi(\mathbf{z})\}$, where $\mathbf{z}$ is a random vector. In this setup, the reconstruction $\hat{\mathbf{x}}$ is constrained to be the output of the DIP network $g_\phi$, and the optimization is performed over both the network parameters $\phi$ and the reconstruction $\mathbf{x}$. Similarly, in Definition 1, $\hat{\mathbf{x}}_0'$ is required to be the output of the posterior mean estimator $f$, which is defined by the network $\boldsymbol{\epsilon}_\theta$.

The subset of images that are in the range of the function $f$ (i.e., backward-consistent) is denoted by $\mathcal{C}_t$ and defined as

$$\mathcal{C}_t := \{f(\mathbf{v}_t; t, \boldsymbol{\epsilon}_\theta) : \mathbf{v}_t \in \mathbb{R}^n\}. \tag{9}$$

Enforcing $\hat{\mathbf{x}}_0'$ to be both measurement- and backward-consistent involves solving the following optimization problem

$$\hat{\mathbf{x}}_0', \hat{\mathbf{v}}_t := \underset{\mathbf{v}_t', \mathbf{x}_0'}{\arg\min} \left\{ \|\mathcal{A}(\mathbf{x}_0') - \mathbf{y}\|_2^2 \quad \text{subject to} \quad \mathbf{x}_0' = f(\mathbf{v}_t'; t, \boldsymbol{\epsilon}_\theta) \right\}. \tag{10}$$

However, Equation (10) may violate forward consistency, as $\hat{\mathbf{v}}_t$ could possibly be far from $\mathbf{x}_t$. Therefore, we propose adding a regularization term, for which Equation (10) becomes

$$\hat{\mathbf{x}}_0', \hat{\mathbf{v}}_t := \underset{\mathbf{v}_t', \mathbf{x}_0'}{\arg\min} \left\{ \|\mathcal{A}(\mathbf{x}_0') - \mathbf{y}\|_2^2 + \lambda \|\mathbf{x}_t - \mathbf{v}_t'\|_2^2 \quad \text{subject to} \quad \mathbf{x}_0' = f(\mathbf{v}_t'; t, \boldsymbol{\epsilon}_\theta) \right\}. \tag{11}$$

During the reverse sampling process, at each time $t$, with the given $\mathbf{x}_t$, we seek a $\mathbf{v}_t'$ in the nearby region (i.e., $\|\mathbf{x}_t - \mathbf{v}_t'\|$ is small), such that $\mathbf{v}_t'$ can be denoised by $f$ to produce a clean image $\mathbf{x}_0'$ (i.e., $\mathbf{x}_0' = f(\mathbf{v}_t'; t, \boldsymbol{\epsilon}_\theta)$), which is also consistent with the measurements $\mathbf{y}$ (i.e., $\|\mathcal{A}(\mathbf{x}_0') - \mathbf{y}\|_2^2$ is small). We need to identify such a $\mathbf{v}_t'$ because $\mathbf{x}_t$ itself cannot be directly denoised by $f$ to yield an

image consistent with the measurements. By substituting the constraint into the objective function, the optimization problem in Equation (11) is reduced to

$$\hat{\mathbf{v}}_t := \arg\min_{\mathbf{v}'_t} \left\{ \|\mathcal{A}\big(f(\mathbf{v}'_t; t, \boldsymbol{\epsilon}_\theta)\big) - \mathbf{y}\|_2^2 + \lambda\|\mathbf{x}_t - \mathbf{v}'_t\|_2^2 \right\}, \quad \hat{\mathbf{x}}'_0 = f(\hat{\mathbf{v}}_t; t, \boldsymbol{\epsilon}_\theta). \qquad (12)$$

The benefit of the considered backward consistency constraint is shown in columns 6 to 8 of Figure 2. After obtaining $\hat{\mathbf{x}}'_0$, the resampling formula in Equation (8) is used to obtain $\mathbf{x}_{t-1}$.

### 3.3 TRIPLE CONSISTENCY CONDITIONS

We now summarize the three key conditions that apply at each sampling step.

(C1) **Measurement Consistency**: The reconstruction $\hat{\mathbf{x}}'_0$ is consistent with the measurements. This means that $\mathcal{A}(\hat{\mathbf{x}}'_0) \approx \mathbf{y}$.

(C2) **Backward Consistency**: The reconstruction $\hat{\mathbf{x}}'_0$ is a denoised image produced by the Tweedie-network denoiser $f$. More generally, we define the backward consistency to include any form of DM network regularization (e.g., using the DM probability-flow (PF) ODE (Karras et al., 2022)) applied to $\hat{\mathbf{x}}'_0$.

(C3) **Forward Consistency**: The pre-trained DM network $\boldsymbol{\epsilon}_\theta$ is provided with in-distribution inputs with high probability. To ensure this, we apply the resampling formula in Equation (8) and enforce that $\hat{\mathbf{v}}_t$ remains close to $\mathbf{x}_t$.

We note that the three considered consistencies are step-wise, meaning they are enforced at every sampling step. This approach contrasts with enforcing these consistencies solely on the final reconstruction at $t = 0$, which represents a significantly weaker requirement.

**C1-C3** aim to ensure that all intermediate reconstructions $\hat{\mathbf{x}}'_0(\mathbf{x}_t)$ (with $t > 0$) are as accurate as possible, allowing us to effectively reduce the number of sampling steps.

Previous works, such as (Song et al., 2023a; Zhang et al., 2024), enforce measurement consistency by applying $\mathcal{A}(\hat{\mathbf{x}}_0) = \mathbf{y}$ exactly, whereas DPS (Chung et al., 2023b) does not ensure consistency along the diffusion trajectory.

### 3.4 THE PROPOSED SAMPLER

Given $\mathbf{x}_t$, $\boldsymbol{\epsilon}_\theta$, and towards satisfying the above conditions, our method, at sampling time $t$, consists of the following three steps:

$$\hat{\mathbf{v}}_t := \arg\min_{\mathbf{v}'_t} \ \|\mathcal{A}\big( \underbrace{\frac{1}{\sqrt{\bar{\alpha}_t}} \big[\mathbf{v}'_t - \sqrt{1-\bar{\alpha}_t}\,\boldsymbol{\epsilon}_\theta(\mathbf{v}'_t, t)\big]}_{f(\mathbf{v}'_t; t, \boldsymbol{\epsilon}_\theta)} \big) - \mathbf{y}\|_2^2 \ + \ \lambda\|\mathbf{x}_t - \mathbf{v}'_t\|_2^2 \qquad (\mathrm{S}_1)$$

$$\hat{\mathbf{x}}'_0 = f(\hat{\mathbf{v}}_t; t, \boldsymbol{\epsilon}_\theta) \equiv \frac{1}{\sqrt{\bar{\alpha}_t}} \big[\hat{\mathbf{v}}_t - \sqrt{1-\bar{\alpha}_t}\,\boldsymbol{\epsilon}_\theta(\hat{\mathbf{v}}_t, t)\big] \qquad (\mathrm{S}_2)$$

$$\mathbf{x}_{t-1} = \sqrt{\bar{\alpha}_{t-1}}\hat{\mathbf{x}}'_0 + \sqrt{1-\bar{\alpha}_{t-1}}\boldsymbol{\eta}_t \ , \ \ \boldsymbol{\eta}_t \sim \mathcal{N}(\mathbf{0}, \mathbf{I}) \ . \qquad (\mathrm{S}_3)$$

The minimization in the first step optimizes over the input $\mathbf{v}'_t$ of the pre-trained diffusion model at time $t$, where the first term of the objective enforces measurement consistency for the posterior mean estimated image, satisfying condition **C1**. The second term serves as a regularization term, implicitly promoting closeness between $\hat{\mathbf{v}}_t$ and $\mathbf{x}_t$ (i.e., condition **C3**), with $\lambda > 0$ acting as the regularization parameter. The argument of the forward operator in Equation ($\mathrm{S}_1$) and the second step in Equation ($\mathrm{S}_2$) enforce that $\hat{\mathbf{v}}_t$ and $\hat{\mathbf{x}}'_0$, respectively, maintain the diffusion trajectory through

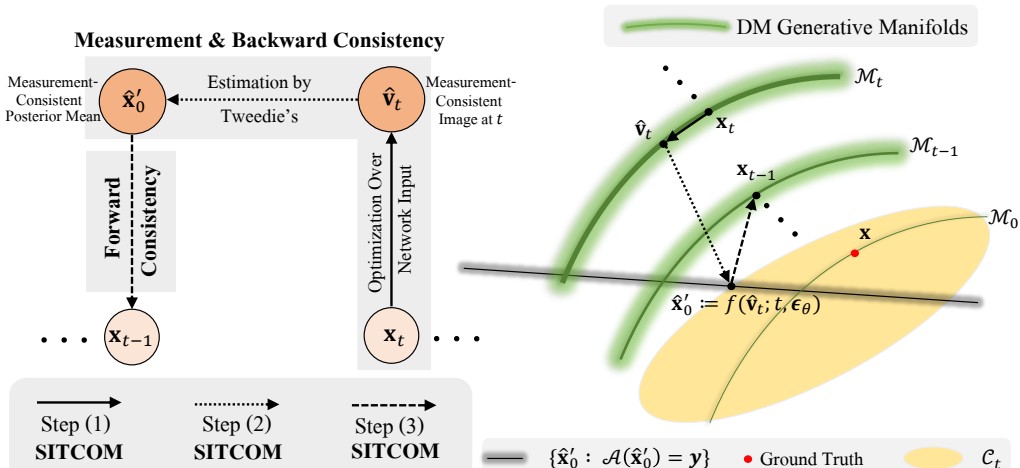

Figure 3: Illustrative diagram of the proposed procedure in SITCOM (*left*). Conceptual illustration of SIT-COM, where $\mathcal{M}_t$ is the DM generative manifold at time $t$ and $\mathcal{C}_t$ is the subset of images that are backward-consistent, defined in Equation (9) (*right*). Step (1) (solid arrow), Step (2) (dotted arrow), and Step (3) (dashed arrow) correspond to Equation ($\text{S}_1$), Equation ($\text{S}_2$), and Equation ($\text{S}_3$), respectively.

obeying Tweedie's formula, thereby satisfying the backward consistency condition, **C2**. After obtaining the measurement-consistent estimate, $\hat{\mathbf{x}}_0'$, as given in Equation ($\text{S}_2$), it must be mapped back to time $t-1$ to generate $\mathbf{x}_{t-1}$. This is achieved through the forward diffusion step in Equation ($\text{S}_3$) as outlined in the forward consistency condition, **C3**. A diagram of SITCOM procedure is provided in Figure 3 (*left*).

**Remark 1.** *Obtaining the estimated image at time 0 given some $\mathbf{x}_t$ using the standard DM PF-ODE (Karras et al., 2022) is more accurate compared to the one-step Tweedie's formula. However, since PF-ODE is an iterative procedure, it requires more computational time. In SITCOM, PF-ODE could replace Tweedie's formula in Equation ($\text{S}_2$). Nevertheless, we chose not to use it, as this would increase the run time, and our empirical results are already highly competitive using Tweedie's formula.*

A conceptual illustration of SITCOM is shown in Figure 3 (*right*). The DM generative manifold, $\mathcal{M}_t$, is defined as the set of all $\mathbf{x}_t$ sampled from $q(\mathbf{x}_t|\mathbf{x}_0) = \mathcal{N}(\mathbf{x}_t; \sqrt{\bar{\alpha}_t}\mathbf{x}_0, (1-\bar{\alpha}_t)\mathbf{I})$, and $\mathbf{x}_0 \sim p_0(\mathbf{x})$. This set coincides with the entire space $\mathbb{R}^n$ equipped with the probability measure induced by the distribution of $\mathbf{x}_t$, which we denote as $\mathcal{P}_t$. In Figure 3 (*right*), the variation of color around each $\mathcal{M}_t$ indicates the concentration of the measure $\mathcal{P}_t$, with darker colors representing higher concentration. SITCOM's Step (1) and Step (2) enforce measurement consistency and backward consistency, thus map $\mathbf{x}_t$ to $\hat{\mathbf{x}}_0' = f(\hat{\mathbf{v}}_t; t, \boldsymbol{\epsilon}_\theta)$ which lies within the intersection of (*i*) measurement-consistent set $\{\hat{\mathbf{x}}_0' : \mathcal{A}(\hat{\mathbf{x}}_0') \approx \mathbf{y}\}$ (the shaded black line) and (*ii*) the backward-consistent set $\mathcal{C}_t$ (the yellow ellipsoid) defined in Equation (9). Subsequently, $\mathbf{x}_{t-1}$ is generated by inserting $\hat{\mathbf{x}}_0'$ into the resampling formula, which enforces the forward consistency.

**Handling Measurement Noise:** To avoid the case where the first term of the objective in Equation ($\text{S}_1$) reaches small values yielding noise overfitting (i.e., when additive Gaussian noise in Equation (1) is considered, $\sigma_{\mathbf{y}} > 0$), we propose refraining from enforcing strict measurement fitting $\mathcal{A}(\mathbf{x}) = \mathbf{y}$. Instead, we use the stopping criterion $\left\| \mathcal{A}\left(\frac{1}{\sqrt{\bar{\alpha}_t}}\left[\mathbf{v}_t' - \sqrt{1-\bar{\alpha}_t}\,\boldsymbol{\epsilon}_\theta(\mathbf{v}_t', t)\right]\right) - \mathbf{y} \right\|_2^2 < \delta^2$, where $\delta \in \mathbb{R}_+$ is a hyper-parameter that indicates the level of tolerance for noise and helps prevent overfitting. This is equivalent to enforcing an $\ell_2$ constraint, and is in spirit similar to (Wang et al., 2024). Since the noise level cannot be accurately estimated, in our experiments, we use $\delta$ that is slightly larger than the actual level of noise in the measurements, i.e., $\delta > \sigma_{\mathbf{y}}\sqrt{m}$.

### 3.5 SITCOM WITH ARBITRARY STEP SIZES

In this subsection, we explain how to apply SITCOM with a large step size and present the final algorithm. The pre-trained DM is trained with $T$ diffusion steps. Given that our method is designed to satisfy measurement and diffusion consistency, SITCOM requires $N \ll T$ sampling iterations, using a step size of $\Delta t := \lfloor \frac{T}{N} \rfloor$. Thus, we introduce the index $i$ instead of $t$ with a relation $t = i\Delta t$.

---

**Algorithm 1 S**tep-**wi**se **T**riple-**Co**nsistent Sa**m**pling (**SITCOM**).

---

**Input**: Measurements $\mathbf{y}$, forward operator $\mathcal{A}(\cdot)$, pre-trained DM $\boldsymbol{\epsilon}_\theta(\cdot\,,\cdot)$, number of diffusion steps $N$, DM noise schedule $\bar{\alpha}_i$ for $i \in \{1, \ldots, N\}$, number of gradient updates $K$, stopping criterion $\delta$, learning rate $\gamma$, and regularization parameter $\lambda$.

 **Output**: Restored image $\hat{\mathbf{x}}$.

 **Initialization**: $\mathbf{x}_N \sim \mathcal{N}(\mathbf{0}, \mathbf{I})$, $\Delta t = \lfloor \frac{T}{N} \rfloor$

 1: **For each** $i \in \{N, N-1, \ldots, 1\}$. (Reducing diffusion sampling steps)

 2:    **Initialize** $\mathbf{v}_i^{(0)} \leftarrow \mathbf{x}_i$. (Initialization to ensure Closeness: **C3** )

 3:    **For each** $k \in \{1, \ldots, K\}$. (Gradient updates for measurement & backward consistency: **C1**, **C2**)

 4:      $\mathbf{v}_i^{(k)} = \mathbf{v}_i^{(k-1)} - \gamma \nabla_{\mathbf{v}_i} \left[ \left\| \mathcal{A}\left( \frac{1}{\sqrt{\bar{\alpha}_i}} \left[ \mathbf{v}_i - \sqrt{1 - \bar{\alpha}_i}\, \boldsymbol{\epsilon}_\theta(\mathbf{v}_i, i\Delta t) \right] \right) - \mathbf{y} \right\|_2^2 + \lambda \|\mathbf{x}_i - \mathbf{v}_i\|_2^2 \right] \Big|_{\mathbf{v}_i = \mathbf{v}_i^{(k-1)}}$.

 5:        **If** $\left\| \mathcal{A}\left( \frac{1}{\sqrt{\bar{\alpha}_i}} \left[ \mathbf{v}_i^{(k)} - \sqrt{1 - \bar{\alpha}_i}\, \boldsymbol{\epsilon}_\theta(\mathbf{v}_i^{(k)}, i\Delta t) \right] \right) - \mathbf{y} \right\|_2^2 < \delta^2$ . (Stopping criterion)

 6:          **Break** the **For** loop in step 3. (Preventing noise overfitting)

 7:    **Assign** $\hat{\mathbf{v}}_i \leftarrow \mathbf{v}_i^{(k)}$. (Backward diffusion consistency of $\hat{\mathbf{v}}_i$: **C2**)

 8:    **Obtain** $\hat{\mathbf{x}}_0' = f(\hat{\mathbf{v}}_i; t, \theta) = \frac{1}{\sqrt{\bar{\alpha}_i}} \left[ \hat{\mathbf{v}}_i - \sqrt{1 - \bar{\alpha}_i}\, \boldsymbol{\epsilon}_\theta(\hat{\mathbf{v}}_i, i\Delta t) \right]$. (Backward consistency of $\hat{\mathbf{x}}_0'$: **C2**)

 9:    **Obtain** $\mathbf{x}_{i-1} = \sqrt{\bar{\alpha}_{i-1}}\hat{\mathbf{x}}_0' + \sqrt{1 - \bar{\alpha}_{i-1}}\boldsymbol{\eta}_i$, $\boldsymbol{\eta}_i \sim \mathcal{N}(\mathbf{0}, \mathbf{I})$ . (Forward diffusion consistency: **C3**)

10: **Restored image:** $\hat{\mathbf{x}} = \mathbf{x}_0$.

---

The procedure of SITCOM is outlined in Algorithm 1. As inputs, SITCOM takes $\mathbf{y}$, $\mathcal{A}(\cdot)$, $\boldsymbol{\epsilon}_\theta$, the number of sampling steps $N$, $\bar{\alpha}_i$ for all $i \in \{1, \ldots, N\}$, the number of optimization steps $K$ per sampling step, stopping criteria $\delta$, and the learning rate $\gamma$.

Starting with initializing $\mathbf{v}_i^{(0)}$ as $\mathbf{x}_i$ (satisfying condition **C3**), lines 3 through 6 correspond to the first step of SITCOM, where Equation ($\mathrm{S}_1$) is solved via either gradient descent (as shown in the algorithm), or the ADAM optimizer (Kingma and Ba, 2015). In lines 5 and 6, the stopping criterion is applied to prevent strict data fidelity (avoiding noise overfitting). Following the gradient updates in the inner loop, $\hat{\mathbf{v}}_i$ is obtained in line 7, which is then used in line 8 to obtain $\hat{\mathbf{x}}_0'$ as specified in Equation ($\mathrm{S}_2$), satisfying condition **C2**. Note that line 8 requires no additional computation, as the $\hat{\mathbf{x}}_0'$ calculated here was already obtained while checking the stopping condition in line 6. After obtaining the double-consistent $\hat{\mathbf{x}}_0'$, the resampling is applied to map the image back to time $t-1$ while ensuring $\mathbf{x}_{t-1}$ to be in-distribution, as indicated in line 9 of the algorithm. In the next iteration, the requirement that $\hat{\mathbf{v}}_{t-1}$ is close to $\mathbf{x}_{t-1}$ ensures that the input $\hat{\mathbf{v}}_{t-1}$ to the DM network, $\boldsymbol{\epsilon}_\theta$, is also in-distribution, thus satisfying the forward-consistency (condition **C3**).

The computational requirements of SITCOM are determined by (*i*) the number of sampling steps $N$ and (*ii*) the number of gradient steps $K$ required for each sampling iteration. Given the proposed stopping criterion, this results in at most $NK$ Number of Function Evaluations (NFEs) of the pre-trained model (forward pass), $NK$ backward passes through the pre-trained model, and $NK$ applications each for the forward operator and its adjoint to solve the optimization problem in Equation ($\mathrm{S}_1$). With early stopping, the computational cost is lower. For example, for a linear operator $\mathcal{A}$ with dimensions $m \times n$, the cost of applying it (or its adjoint) to a vector is $\mathcal{O}(mn)$. For a network with width $M$ and depth $L$, the cost for making a forward pass is $\mathcal{O}(LM^2)$. The gradients are computed w.r.t. the input of the DM network, requiring an additional backward pass. Consequently, this procedure is significantly more efficient than network training, where the network weights are updated instead of the input.

## 3.6 RELATION WITH EXISTING APPROACHES

While SITCOM and DPS (Chung et al., 2023b) both use Tweedie's formula, there are two major differences. First, DPS does not enforce backward consistency. Specifically, it only considers one gradient descent step of the optimization in Equation ($\mathrm{S}_1$), whereas our method perform multiple steps, initializing with $\mathbf{x}_t$. Second, DPS does not enforce the forward diffusion consistency, namely, it does not use resampling Equation ($\mathrm{S}_3$). This means that DPS does not enforce a step-wise **C1**-**C3**. Both SITCOM and the works in (Song et al., 2023a; Zhang et al., 2024) are optimization-based methods that modify the sampling steps to enforce measurement consistency, and both involve

mapping back to time $t-1$ (as in step 3 of SITCOM). However, there is a major difference between them: The optimization variable in these works is the estimated image at time $t$ (the output of the DM network), whereas in SITCOM, it is the noisy image at time $t$ (the input of the network). This means that these studies enforce **C1** and **C3**, but not **C2**.

It is worth noting that while some previous works, such as RED-diff (Wang et al., 2024) and DMPlug (Mardani et al., 2023), also utilize the implicit bias of the network, they adopt the full diffusion process as a regularizer, applied only once. In contrast, our method uses the neural network as the regularizer at each iteration and focuses specifically on reducing the number of sampling steps for a given level of accuracy.

## 4    EXPERIMENTAL RESULTS

**Tasks, Baselines, & Datasets:**    Our experimental setup for IPs and noise levels used largely follows DPS (Chung et al., 2023b). For linear IPs, we evaluate five tasks: super resolution, Gaussian deblurring, motion deblurring, box inpainting, and random inpainting. For Gaussian deblurring and motion deblurring, we use $61\times61$ kernels with standard deviations of 3 and 0.5, respectively. In the super-resolution task, a bicubic resizer downscales images by a factor of 4. For box inpainting, a random $128\times128$ box is applied to mask image pixels, and for random inpainting, the mask is generated with each pixel masked with a probability of 0.7, as described in (Song et al., 2023a). For nonlinear IP tasks, we consider three tasks: phase retrieval, high dynamic range (HDR) reconstruction, and nonlinear (non-uniform) deblurring. For phase retrieval, an oversampling rate of 2 is applied in frequency domain, and we report the best result out of four independent samples, consistent with (Chung et al., 2023b; Zhang et al., 2024) (see Appendix D for more discussion on phase retrieval). In HDR reconstruction, the goal is to restore a higher dynamic range image from a lower dynamic range image (with a factor of 2). Nonlinear deblurring follows the setup in (Tran et al., 2021). For measurement noise, we use $\sigma_{\mathbf{y}} \in \{0.01, 0.05\}$ for all tasks. For baselines, in this section, we use DPS (Chung et al., 2023b), DDNM (Wang et al., 2022), DCDP (Li et al., 2024), and DAPS (Zhang et al., 2024). The selection criteria is based on these baselines' competitive performance on several linear and non-linear inverse problems under measurement noise. Additionally, we provide comparison results with three other baselines in Table 5 of Appendix E. We evaluate SITCOM and baselines using 100 test images from the validation set of FFHQ (Karras et al., 2019) and 100 test images from the validation set of ImageNet (Deng et al., 2009) for which the FFHQ-trained and ImageNet-trained DMs are given in (Chung et al., 2023b) and (Dhariwal and Nichol, 2021), respectively, following the previous convention. For evaluation metrics, we use PSNR, SSIM (Wang et al., 2004), and LPIPS (Zhang et al., 2018).

**SITCOM Settings:**    For Algorithm 1, we set $N = 20$ and $K = 30$ for most tasks. We show the impact of $N$ and $K$ in Appendix F.1. The parameter $\lambda$ is set to 0 for all tasks other than phase retrieval where we use $\lambda = 1$, following the ablation study in Appendix F.2. The impact of the stopping criterion under the noisy setting is given in Appendix F.3. The learning rate for Equation ($\mathtt{S_1}$) is set to $\gamma = 0.01$ across all measurements noise levels, datasets, and tasks. Table 8 in Appendix F.4 lists all the hyper-parameters used for every task. We note that the exact set of hyper-parameters is used for the FFHQ and ImageNet datasets. Our code is available online[2].

**Main Results:**    In Table 1, we present the quantitative results in terms of the average PSNR, SSIM, LPIPS, and run-time (minutes). Columns 3 to 6 correspond to the FFHQ dataset, while columns 7 to 10 reflect results for the ImageNet dataset. The table covers 8 tasks, 4 evaluation metrics, and 2 datasets, totaling 64 results. Among these, SITCOM reports the best performance in 58 out of 64 cases. On average, SITCOM demonstrates strong reconstruction capabilities across most tasks. For the FFHQ dataset, SITCOM reports a PSNR improvement of over 1 dB in Super Resolution, random In-painting, and Gaussian Deblurring compared to the second-best method. On ImageNet, we observe more than a 1 dB improvement in random In-painting. Other than ImageNet Gaussian Deblurring and ImageNet Phase Retrieval, for which we under-perform by 0.66 dB and 0.31 dB, respectively, our PSNR improvement when compared to the second-best results are less than 1 dB. However, in terms of run-time, SITCOM consistently requires less computational time across all tasks. For FFHQ, SITCOM is over $3\times$ faster in Box In-painting and motion Deblurring, and more

---

[2]https://anonymous.4open.science/r/SITCOM-7539/README.md

| Task | Method | FFHQ | | | | ImageNet | | | |
|---|---|---|---|---|---|---|---|---|---|
| | | PSNR (↑) | SSIM (↑) | LPIPS (↓) | Run-time (↓) | PSNR (↑) | SSIM (↑) | LPIPS (↓) | Run-time (↓) |
| Super Resolution 4× | DPS | $24.44_{\pm0.56}$ | $0.801_{\pm0.032}$ | $0.26_{\pm0.022}$ | $1.26_{\pm0.52}$ | $23.86_{\pm0.34}$ | $0.76_{\pm0.041}$ | $0.357_{\pm0.069}$ | $2.38_{\pm1.02}$ |
| | DAPS | $29.24_{\pm0.42}$ | $0.851_{\pm0.024}$ | $0.135_{\pm0.039}$ | $1.24_{\pm0.22}$ | $25.67_{\pm0.73}$ | $0.802_{\pm0.045}$ | $0.256_{\pm0.067}$ | $2.16_{\pm0.45}$ |
| | DDNM | $28.02_{\pm0.78}$ | $0.842_{\pm0.034}$ | $0.197_{\pm0.034}$ | $1.07_{\pm0.42}$ | $23.96_{\pm0.89}$ | $0.767_{\pm0.045}$ | $0.475_{\pm0.044}$ | $1.27_{\pm0.55}$ |
| | DCDP | $27.88_{\pm1.34}$ | $0.825_{\pm0.07}$ | $0.211_{\pm0.05}$ | $0.52_{\pm0.34}$ | $24.12_{\pm1.24}$ | $0.772_{\pm0.000}$ | $0.351_{\pm0.00}$ | $1.45_{\pm0.00}$ |
| | SITCOM (ours) | $30.68_{\pm1.02}$ | $0.867_{\pm0.045}$ | $0.142_{\pm0.056}$ | $0.45_{\pm0.58}$ | $26.35_{\pm1.21}$ | $0.812_{\pm0.021}$ | $0.232_{\pm0.038}$ | $1.12_{\pm0.52}$ |
| Box In-Painting | DPS | $23.20_{\pm0.89}$ | $0.754_{\pm0.023}$ | $0.196_{\pm0.032}$ | $1.57_{\pm0.55}$ | $19.78_{\pm0.78}$ | $0.691_{\pm0.052}$ | $0.312_{\pm0.025}$ | $2.28_{\pm1.02}$ |
| | DAPS | $24.17_{\pm1.02}$ | $0.787_{\pm0.032}$ | $0.135_{\pm0.032}$ | $1.35_{\pm0.45}$ | $21.43_{\pm0.40}$ | $0.736_{\pm0.020}$ | $0.218_{\pm0.021}$ | $2.54_{\pm1.02}$ |
| | DDNM | $24.37_{\pm0.45}$ | $0.792_{\pm0.024}$ | $0.232_{\pm0.026}$ | $1.02_{\pm0.032}$ | $21.64_{\pm0.66}$ | $0.732_{\pm0.028}$ | $0.319_{\pm0.015}$ | $1.45_{\pm1.02}$ |
| | DCDP | $23.66_{\pm1.67}$ | $0.762_{\pm0.07}$ | $0.144_{\pm0.05}$ | $0.56_{\pm0.45}$ | $20.45_{\pm1.22}$ | $0.712_{\pm0.07}$ | $0.298_{\pm0.04}$ | $1.127_{\pm0.25}$ |
| | SITCOM (ours) | $24.68_{\pm0.78}$ | $0.801_{\pm0.042}$ | $0.121_{\pm0.08}$ | $0.35_{\pm0.25}$ | $21.88_{\pm0.92}$ | $0.742_{\pm0.032}$ | $0.214_{\pm0.021}$ | $1.12_{\pm0.35}$ |
| Random In-Painting | DPS | $28.39_{\pm0.82}$ | $0.844_{\pm0.042}$ | $0.194_{\pm0.021}$ | $1.52_{\pm0.30}$ | $24.26_{\pm0.42}$ | $0.772_{\pm0.02}$ | $0.326_{\pm0.034}$ | $2.27_{\pm0.25}$ |
| | DAPS | $31.02_{\pm0.45}$ | $0.902_{\pm0.015}$ | $0.098_{\pm0.017}$ | $1.56_{\pm0.40}$ | $28.44_{\pm0.45}$ | $0.872_{\pm0.024}$ | $0.135_{\pm0.052}$ | $2.14_{\pm0.45}$ |
| | DDNM | $29.93_{\pm0.67}$ | $0.889_{\pm0.042}$ | $0.122_{\pm0.034}$ | $1.45_{\pm0.35}$ | $29.22_{\pm0.55}$ | $0.912_{\pm0.034}$ | $0.191_{\pm0.048}$ | $1.54_{\pm0.52}$ |
| | DCDP | $28.59_{\pm0.95}$ | $0.852_{\pm0.06}$ | $0.202_{\pm0.04}$ | $0.55_{\pm0.25}$ | $26.22_{\pm1.13}$ | $0.791_{\pm0.06}$ | $0.289_{\pm0.03}$ | $1.44_{\pm0.34}$ |
| | SITCOM (ours) | $32.05_{\pm1.02}$ | $0.909_{\pm0.09}$ | $0.095_{\pm0.025}$ | $0.45_{\pm0.50}$ | $29.60_{\pm0.78}$ | $0.915_{\pm0.028}$ | $0.127_{\pm0.039}$ | $1.14_{\pm0.45}$ |
| Gaussian Deblurring | DPS | $25.52_{\pm0.78}$ | $0.826_{\pm0.052}$ | $0.211_{\pm0.017}$ | $1.50_{\pm0.50}$ | $21.86_{\pm0.45}$ | $0.772_{\pm0.08}$ | $0.362_{\pm0.034}$ | $2.55_{\pm0.45}$ |
| | DAPS | $29.22_{\pm0.50}$ | $0.884_{\pm0.056}$ | $0.164_{\pm0.032}$ | $1.40_{\pm0.52}$ | $26.12_{\pm0.78}$ | $0.832_{\pm0.092}$ | $0.245_{\pm0.022}$ | $2.23_{\pm0.52}$ |
| | DDNM | $28.22_{\pm0.52}$ | $0.867_{\pm0.056}$ | $0.216_{\pm0.042}$ | $1.56_{\pm0.45}$ | $28.06_{\pm0.52}$ | $0.879_{\pm0.072}$ | $0.278_{\pm0.089}$ | $1.75_{\pm0.63}$ |
| | DCDP | $26.67_{\pm0.78}$ | $0.835_{\pm0.08}$ | $0.196_{\pm0.04}$ | $0.56_{\pm0.25}$ | $23.24_{\pm1.18}$ | $0.781_{\pm0.06}$ | $0.343_{\pm0.04}$ | $1.34_{\pm0.43}$ |
| | SITCOM (ours) | $30.25_{\pm0.89}$ | $0.892_{\pm0.032}$ | $0.135_{\pm0.078}$ | $0.46_{\pm0.25}$ | $27.40_{\pm0.45}$ | $0.854_{\pm0.045}$ | $0.236_{\pm0.039}$ | $1.10_{\pm0.42}$ |
| Motion Deblurring | DPS | $23.40_{\pm1.42}$ | $0.737_{\pm0.024}$ | $0.270_{\pm0.025}$ | $2.40_{\pm0.55}$ | $21.86_{\pm2.05}$ | $0.724_{\pm0.022}$ | $0.357_{\pm0.032}$ | $2.56_{\pm0.40}$ |
| | DAPS | $29.66_{\pm0.50}$ | $0.872_{\pm0.027}$ | $0.157_{\pm0.012}$ | $1.86_{\pm0.12}$ | $27.86_{\pm1.20}$ | $0.862_{\pm0.032}$ | $0.196_{\pm0.021}$ | $2.3_{\pm0.45}$ |
| | SITCOM (ours) | $30.34_{\pm0.67}$ | $0.902_{\pm0.037}$ | $0.148_{\pm0.041}$ | $0.5_{\pm0.45}$ | $28.65_{\pm0.34}$ | $0.876_{\pm0.021}$ | $0.189_{\pm0.036}$ | $1.48_{\pm0.35}$ |
| Phase Retrieval | DPS | $17.34_{\pm2.67}$ | $0.67_{\pm0.045}$ | $0.41_{\pm0.08}$ | $1.50_{\pm0.34}$ | $16.82_{\pm1.22}$ | $0.64_{\pm0.08}$ | $0.447_{\pm0.032}$ | $2.17_{\pm0.24}$ |
| | DAPS | $30.67_{\pm3.12}$ | $0.908_{\pm0.041}$ | $0.122_{\pm0.084}$ | $1.34_{\pm0.78}$ | $25.76_{\pm2.33}$ | $0.797_{\pm0.045}$ | $0.255_{\pm0.095}$ | $2.24_{\pm0.25}$ |
| | DCDP | $28.52_{\pm2.50}$ | $0.892_{\pm0.19}$ | $0.167_{\pm0.92}$ | $3.30_{\pm0.45}$ | $24.25_{\pm2.25}$ | $0.778_{\pm0.14}$ | $0.287_{\pm0.089}$ | $3.49_{\pm0.52}$ |
| | SITCOM (ours) | $30.97_{\pm3.10}$ | $0.915_{\pm0.064}$ | $0.112_{\pm0.102}$ | $0.52_{\pm0.34}$ | $25.45_{\pm2.78}$ | $0.808_{\pm0.065}$ | $0.246_{\pm0.088}$ | $1.40_{\pm0.40}$ |
| Non-Uniform Deblurring | DPS | $23.42_{\pm2.15}$ | $0.757_{\pm0.042}$ | $0.279_{\pm0.067}$ | $1.55_{\pm0.44}$ | $22.57_{\pm0.67}$ | $0.778_{\pm0.067}$ | $0.310_{\pm0.102}$ | $2.35_{\pm0.45}$ |
| | DAPS | $28.23_{\pm1.55}$ | $0.833_{\pm0.052}$ | $0.155_{\pm0.041}$ | $1.42_{\pm0.41}$ | $27.65_{\pm1.2}$ | $0.822_{\pm0.056}$ | $0.169_{\pm0.044}$ | $2.14_{\pm0.45}$ |
| | DCDP | $28.78_{\pm1.44}$ | $0.827_{\pm0.08}$ | $0.162_{\pm0.04}$ | $3.30_{\pm0.45}$ | $26.56_{\pm1.09}$ | $0.803_{\pm0.06}$ | $0.182_{\pm0.05}$ | $3.70_{\pm0.36}$ |
| | SITCOM (ours) | $30.12_{\pm0.68}$ | $0.902_{\pm0.042}$ | $0.145_{\pm0.037}$ | $0.52_{\pm0.45}$ | $28.78_{\pm0.79}$ | $0.832_{\pm0.056}$ | $0.16_{\pm0.048}$ | $1.25_{\pm0.45}$ |
| High Dynamic Range | DPS | $22.88_{\pm1.25}$ | $0.722_{\pm0.056}$ | $0.264_{\pm0.089}$ | $1.45_{\pm0.34}$ | $19.33_{\pm1.45}$ | $0.688_{\pm0.067}$ | $0.503_{\pm0.132}$ | $2.42_{\pm0.46}$ |
| | DAPS | $27.12_{\pm0.89}$ | $0.825_{\pm0.056}$ | $0.166_{\pm0.078}$ | $1.25_{\pm0.35}$ | $26.30_{\pm1.02}$ | $0.792_{\pm0.046}$ | $0.177_{\pm0.089}$ | $2.18_{\pm0.55}$ |
| | SITCOM (ours) | $27.98_{\pm1.06}$ | $0.832_{\pm0.052}$ | $0.158_{\pm0.032}$ | $0.52_{\pm0.30}$ | $26.97_{\pm0.87}$ | $0.821_{\pm0.045}$ | $0.167_{\pm0.052}$ | $1.54_{\pm0.35}$ |

Table 1: Average PSNR, SSIM, LPIPS, and run-time (minutes) of SITCOM and baselines using 100 test images from the **FFHQ** dataset (columns 3 to 7) and 100 test images from the **ImageNet** dataset with a **measurement noise level of** $\sigma_{\mathbf{y}} = 0.05$. The results for the $\sigma_{\mathbf{y}} = 0.01$ case are given in Table 4 of Appendix E. The first five tasks are linear, while the last three tasks are non-linear (underlined). For each task and dataset combination, the best results are bolded, and the second-best results are underlined. Values after $\pm$ represent the standard deviation. All results were obtained using a **single RTX5000 GPU** machine. For phase retrieval, the run-time is reported for the best result out of four independent runs. This is applied for SITCOM and baselines. More discussion about phase retrieval is given in Appendix D.

than $2\times$ faster in the remaining tasks, whereas on ImageNet, the run-time improvement ranges from 36 seconds (for HDR) to 62.4 seconds (for Super Resolution), when compared to DPS, DDNM, and DAPS. For linear tasks, SITCOM requires slightly less run-time than DCDP on both datasets. However, across the two datasets, SITCOM achieves PSNR improvements of more than 1 dB, 2 dB, and 3 dB for the tasks of super resolution, box in-painting, and random in-painting (and Gaussian Deblurring), respectively, as compared to DCDP. For non-linear tasks, SITCOM not only provides PSNR improvements over DCDP but also significantly reduces run-time.

In summary, the results in Table 1 demonstrate that SITCOM either provides a notable improvement in restoration quality (e.g., cases where we report PSNR improvements of over 1 dB) or delivers comparable results to the baselines, all while reducing computation time. In Appendix E, we present the results with $\sigma_{\mathbf{y}} = 0.01$ case (Table 4). Additionally, Table 5 includes quantitative results for three more baselines. In addition to the FFHQ restored images in Figure 1, we also provide additional samples from both datasets in the figures found in Appendix H.

## 5 CONCLUSION

In this paper, we proposed three conditions to achieve measurement- and diffusion-consistent trajectories for linear and non-linear inverse imaging problems using diffusion models (DMs) as priors. These conditions form the basis of our unique optimization-based sampling method, which optimizes the input of the diffusion model at each step. This approach allows for greater control over the diffusion process and enhances data consistency with the given measurements. Through extensive experiments across eight image restoration tasks, we evaluated the effectiveness of our method. The results showed that our sampler consistently delivers improved or comparable quantitative performance against state-of-the-art baselines, even with measurement noise. Notably, our method is efficient, requiring significantly less run-time than leading baselines, making it practical for real-world applications.

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

# Appendix

In the Appendix, we start by showing the equivalence between the second formula in Equation (3) and Equation (6) (Appendix A). Then, we discuss the known limitations and future extensions of SITCOM (Appendix B). Subsequently, we present experiments to highlight the impact of the proposed backward consistency (Appendix C). This is followed by a discussion on phase retrieval (Appendix D). In Appendix E, we provide further comparison results, and in Appendix F, we perform ablation studies to examine the effects of the stopping criterion and other components/hyper-parameters in SITCOM. Appendix G covers the implementation details of tasks and baselines, followed by examples of restored images (Appendix H).

## A    DERIVATION OF EQUATION (6)

From (Luo, 2022), we have

$$\mathbf{s}_\theta(\mathbf{x}_t, t) = -\frac{1}{\sqrt{1 - \bar{\alpha}_t}} \boldsymbol{\epsilon}_\theta(\mathbf{x}_t, t) \, . \tag{13}$$

Rearranging the Tweedie's formula in Equation (4) to solve for $\boldsymbol{\epsilon}_\theta(\mathbf{x}_t, t)$ yields

$$\boldsymbol{\epsilon}_\theta(\mathbf{x}_t, t) = \frac{\mathbf{x}_t - \sqrt{\bar{\alpha}_t}\hat{\mathbf{x}}_0(\mathbf{x}_t)}{\sqrt{1 - \bar{\alpha}_t}} \, . \tag{14}$$

Now, we substitute into the recursive equation for $\mathbf{x}_{t-1}$:

$$\mathbf{x}_{t-1} = \frac{1}{\sqrt{1 - \beta_t}} \left[ \mathbf{x}_t + \beta_t \mathbf{s}_\theta(\mathbf{x}_t, t) \right] + \sqrt{\beta_t}\boldsymbol{\eta}_t \tag{15}$$

$$= \frac{1}{\sqrt{1 - \beta_t}} \left[ \mathbf{x}_t + \beta_t \left( -\frac{1}{\sqrt{1 - \bar{\alpha}_t}} \boldsymbol{\epsilon}_\theta(\mathbf{x}_t, t) \right) \right] + \sqrt{\beta_t}\boldsymbol{\eta}_t \tag{16}$$

$$= \frac{1}{\sqrt{1 - \beta_t}} \left[ \mathbf{x}_t - \frac{\beta_t}{\sqrt{1 - \bar{\alpha}_t}} \boldsymbol{\epsilon}_\theta(\mathbf{x}_t, t) \right] + \sqrt{\beta_t}\boldsymbol{\eta}_t \tag{17}$$

$$= \frac{1}{\sqrt{1 - \beta_t}} \left[ \mathbf{x}_t - \frac{\beta_t}{\sqrt{1 - \bar{\alpha}_t}} \left( \frac{\mathbf{x}_t - \sqrt{\bar{\alpha}_t}\hat{\mathbf{x}}_0(\mathbf{x}_t)}{\sqrt{1 - \bar{\alpha}_t}} \right) \right] + \sqrt{\beta_t}\boldsymbol{\eta}_t \tag{18}$$

$$= \frac{1}{\sqrt{1 - \beta_t}} \left[ \mathbf{x}_t - \frac{\beta_t}{1 - \bar{\alpha}_t} \left( \mathbf{x}_t - \sqrt{\bar{\alpha}_t}\hat{\mathbf{x}}_0(\mathbf{x}_t) \right) \right] + \sqrt{\beta_t}\boldsymbol{\eta}_t \tag{19}$$

$$= \frac{1}{\sqrt{1 - \beta_t}} \left[ \left( 1 - \frac{\beta_t}{1 - \bar{\alpha}_t} \right) \mathbf{x}_t + \frac{\sqrt{\bar{\alpha}_t}\beta_t}{1 - \bar{\alpha}_t} \hat{\mathbf{x}}_0(\mathbf{x}_t) \right] + \sqrt{\beta_t}\boldsymbol{\eta}_t \tag{20}$$

$$= \frac{(1 - \bar{\alpha}_t - \beta_t)}{\sqrt{1 - \beta_t}\,(1 - \bar{\alpha}_t)} \mathbf{x}_t + \frac{\sqrt{\bar{\alpha}_t}\beta_t}{\sqrt{1 - \beta_t}\,(1 - \bar{\alpha}_t)} \hat{\mathbf{x}}_0(\mathbf{x}_t) + \sqrt{\beta_t}\boldsymbol{\eta}_t \tag{21}$$

$$= \frac{(\alpha_t - \bar{\alpha}_t)}{\sqrt{\alpha_t}\,(1 - \bar{\alpha}_t)} \mathbf{x}_t + \frac{\sqrt{\bar{\alpha}_t}\beta_t}{\sqrt{\alpha_t}\,(1 - \bar{\alpha}_t)} \hat{\mathbf{x}}_0(\mathbf{x}_t) + \sqrt{\beta_t}\boldsymbol{\eta}_t \tag{22}$$

$$= \frac{\left( \sqrt{\alpha_t} - \sqrt{\alpha_t}\bar{\alpha}_{t-1} \right)}{1 - \bar{\alpha}_t} \mathbf{x}_t + \frac{\sqrt{\bar{\alpha}_{t-1}}\beta_t}{1 - \bar{\alpha}_t} \hat{\mathbf{x}}_0(\mathbf{x}_t) + \sqrt{\beta_t}\boldsymbol{\eta}_t \tag{23}$$

$$= \frac{\sqrt{\alpha_t}\,(1 - \bar{\alpha}_{t-1})}{1 - \bar{\alpha}_t} \mathbf{x}_t + \frac{\sqrt{\bar{\alpha}_{t-1}}\beta_t}{1 - \bar{\alpha}_t} \hat{\mathbf{x}}_0(\mathbf{x}_t) + \sqrt{\beta_t}\boldsymbol{\eta}_t \, , \tag{24}$$

which is equivalent to the second formula in Equation (3).

## B    LIMITATIONS & FUTURE WORK

In SITCOM, the stopping criterion parameter is set slightly higher than the level of measurement noise, determined by $\sigma_\mathbf{y}$. As a result, our method requires access to (or estimation of) the measurement noise prior to the restoration process. Knowledge of noise level is also assumed in other works

such as DAPS (Zhang et al., 2024). In practice, classical approaches, such as (Liu et al., 2006; Chen et al., 2015), can be used to estimate the noise.

Additionally, the stated conditions and proposed sampler are limited to the non-blind setting, as SITCOM assumes full access to the forward model, unlike works such as (Chung et al., 2023a), which perform both image restoration and forward model estimation.

For future work, in addition to addressing the aforementioned limitations, we aim to extend SIT-COM to the latent space and explore its applicability in medical image reconstruction.

## C IMPACT OF THE PROPOSED BACKWARD CONSISTENCY

Here, we demonstrate the impact of the proposed backward diffusion consistency in SITCOM using two experiments. We note that, in SITCOM, we apply a step-wise network regularization for the backward consistency such that we fully exploit the implicit regularization of the network. Removing the step-wise network regularization is equivalent to removing the requirement for $\hat{\mathbf{x}}_0 = f(\mathbf{v}_t; \theta, t)$. This makes $\mathbf{x}_0$ a free variable which reduces the optimization in Equation (S1) to Equation (5).

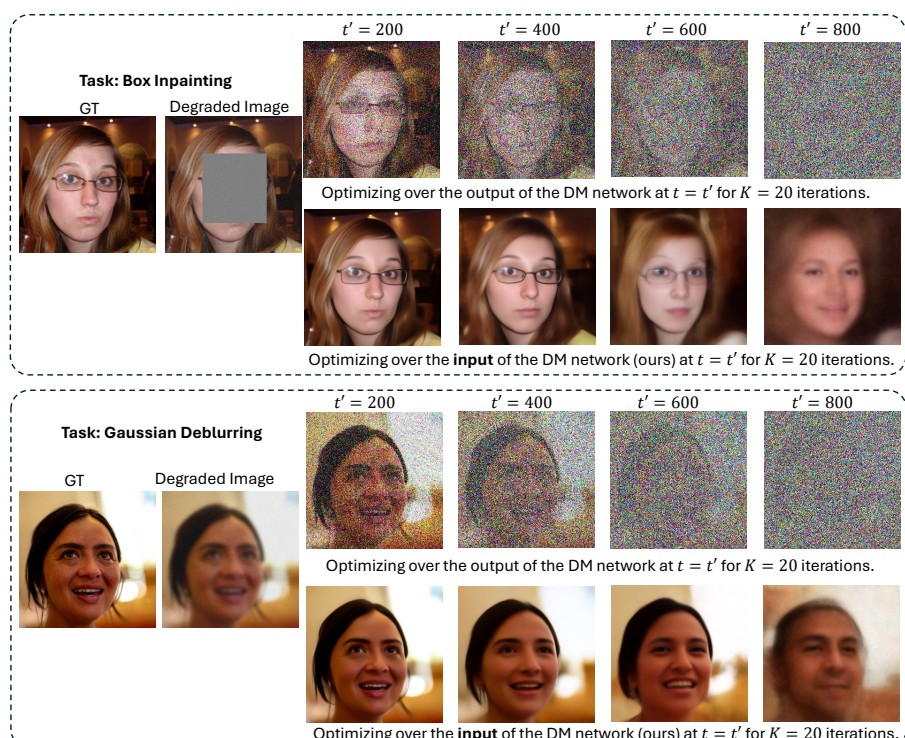

Figure 4: Results of applying optimization-based measurement consistency, for which the optimization variable is the DM output (resp. input), are shown in the first (resp. second) row for each task: Box Inpainting (*top*) and Gaussian Deblurring (*bottom*).

First, for the box-painting task, we compare optimizing over the input to the DM (as in SITCOM) with optimizing over the output of the DM network (as is done in DCDP (Li et al., 2024) and DAPS (Zhang et al., 2024)) at time steps $t' \in \{200, 400, 600\}$. For each case (selection of $t'$), we start from $t = T$ and run SITCOM with a step size of $\lfloor \frac{T}{N} \rfloor$. At $t = t'$, given $\mathbf{x}_{t'}$, we perform two separate optimizations with intializing the optimization variable as $\mathbf{x}_{t'}$: one iteratively over the DM network input (ours) and another iteratively over the DM network output (i.e., Equation (5) but without the regularization), both running until convergence (i.e., when the loss stops decreasing). For our approach, the result of the optimization from Equation (S1) is used as input to Tweedie's formula in Equation (S2) to compute the posterior mean $\hat{\mathbf{x}}'_0 = \hat{\mathbf{x}}_0(\mathbf{v}_t)$. For the case of optimizing over the DM output, we use Equation (5) without regularization. Figure 2 shows the results at different time steps. The consistency between the ground truth and the unmasked regions of the estimated

| Task | Method | $t'$ with PSNR/run-time | | | |
|---|---|---|---|---|---|
| | | 200 | 400 | 600 | 800 |
| Super Resolution 4× | SITCOM at $K = 1000$ | 31.32/558.45 | 30.78/476.09 | 29.89/424.26 | 25.62/256.16 |
| | SITCOM without Backward Consistency at $K = 20$ | 25.24/529.56 | 24.74/452.22 | 17.56/414.45 | 14.45/245.87 |
| Non-linear Deblurring | SITCOM at $K = 1000$ | 30.78/544.45 | 30.56/483.78 | 28.23/424.69 | 26.78/229.24 |
| | SITCOM without Backward Consistency at $K = 1000$ | 24.88/529.44 | 23.56/464.88 | 21.45/412.68 | 12.25/216.47 |

Table 2: Average PSN/run-time results of 20 FFHQ test images at intermediate time steps to show the impact of the proposed step-wise network regularization for backward consistency. Here, we use $K = 1000$ at each $t'$ for both cases.

| Task | Method | $t'$ with PSNR/run-time | | | |
|---|---|---|---|---|---|
| | | 200 | 400 | 600 | 800 |
| Super Resolution 4× | SITCOM at $K = 20$ | 30.22/30.45 | 28.78/26.09 | 25.45/14.26 | 22.22/10.16 |
| | SITCOM without Backward Consistency at $K = 20$ | 15.24/29.56 | 13.24/25.12 | 11.56/14.45 | 9.45/7.87 |
| Non-linear Deblurring | SITCOM at $K = 20$ | 29.78/29.45 | 27.56/23.78 | 24.23/13.69 | 21.78/10.16 |
| | SITCOM without Backward Consistency at $K = 20$ | 14.79/29.44 | 13.56/23.48 | 11.45/12.88 | 8.25/9.47 |

Table 3: Average PSN/run-time results of 20 FFHQ test images at intermediate time steps to show the impact of the proposed step-wise network regularization for backward consistency. Here, we use $K = 20$ at each $t'$ for both cases.

images suggest the convergence of the measurement consistency. As observed, SITCOM produces significantly less artifacts in the masked region when compared to optimizing over the output. This is evident both at earlier time steps ($t' = 600$) and later steps ($t' = 400$ and $t' = 200$).

In Table 2, we repeat the above experiment for super-resolution and Non-linear Deblurring where we report the average PSNR and run-time of 20 FFHQ test images. As observed, given the same number of gradient updates ($K = 1000$), optimizing over the input consistently achieves better results, highlighting the impact of the proposed step-wise network regularization for the backward consistency. We note that the lower PSNRs is due not running the algorithm until convergence.

For the second experiment, the goal is to show that SITCOM requires much smaller number of optimization steps to remove the noise as compared to the case where the optimization variable is the output of the DM network. The results are given in Figure 4, where we repeat the above experiment with two tasks: Box-inpainting (*top*) and Gaussian Deblurring (*bottom*), this time using a fixed number of optimization steps for both SITCOM, and optimizing over the DM output. Specifically, we run SITCOM from $t = T$ to $t = t' + 1$. Then, we apply $K = 20$ iterations (the setting in SITCOM) in Equation ($S_1$), and $K = 20$ when optimizing Equation (5) (without regularization) where measurement noise is $\sigma_{\mathbf{y}} = 0.05$. As shown, compared to optimizing over the DM output, SITCOM significantly reduces noise across all considered $t'$, underscoring the effect of the proposed backward diffusion consistency when optimizing over the DM input.

In Table 3, we repeat the second experiment for two more tasks: super-resolution and Non-linear Deblurring. Similar to Table 2, we report the average PSNR and run-time of 20 FFHQ test images. The results indicate that even at lower $K$, using the proposed backward consistency return the best results.

# D  DISCUSSION ON PHASE RETRIEVAL

As discussed in our experimental results section, for the phase retrieval task, we report the best results from 4 independent runs, following the convention in (Chung et al., 2023b; Zhang et al., 2024). For the phase retrieval results of Table 1 and Table 4 (given in Appendix E), we use this approach across all baselines where the run-time is reported for one run.

The forward model for phase retrieval is adopted from DPS where the inverse problem is generally more challenging compared to other image restoration tasks. This increased difficulty arises from the presence of multiple modes that can yield the same measurements (Zhang et al., 2024).

In Figure 5, we present two examples comparing SITCOM, DPS, and DAPS. For each ground truth image, we show four results from which the best one was selected. In the first column, SITCOM avoids significant artifacts, while DAPS produces one image rotated by 180 degrees. In the second column, both SITCOM and DAPS exhibit one run with severe artifacts. However, the last image from SITCOM does exhibit more artifacts compared to the second worst-case result from DAPS.

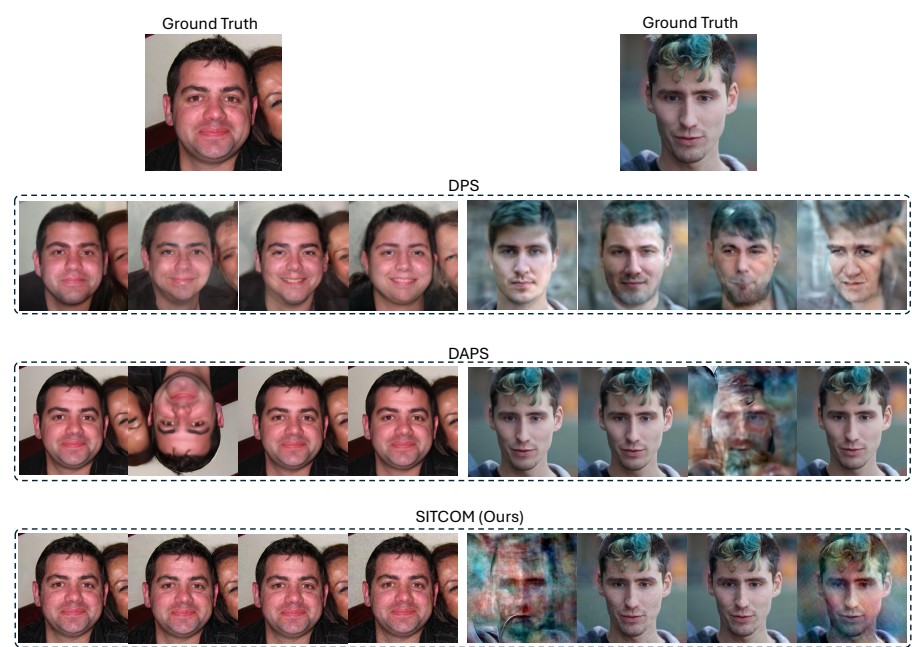

Figure 5: Results of Phase Retrieval on two images (top row) from the FFHQ dataset. Rows 2, 3, and 4 correspond to the results of DPS, DAPS, and SITCOM (ours), respectively.

Additionally, the DPS results show severe perceptual differences in both cases, with artifacts being particularly noticeable in the second column.

# E  ADDITIONAL COMPARISON RESULTS

In Table 4, we present the average PSNR, SSIM, LPIPS, and run-time (minutes) of DPS, DAPS, DDNM, and SITCOM using the FFHQ and ImageNet datasets for which the measurement noise level is set to $\sigma_{\mathbf{y}} = 0.01$ (different from Table 1). The goal of these results is to evaluate our method and baselines under less noisy settings.

Overall, we observe similar trends to those discussed in Section 4 for Table 1. On the FFHQ dataset, SITCOM achieves higher average PSNR values compared to the baselines across all tasks, with improvements exceeding 1 dB in 5 out of 8 tasks. For the ImageNet dataset, we observe more than 1 dB improvement on the non-linear deblurring task, while for the remaining tasks, the improvement is less than 1 dB, except for Gaussian deblurring (where SITCOM underperforms by 0.22 dB) and phase retrieval (underperforming by 0.36 dB).

In terms of run-time, generally, SITCOM significantly outperforms DDNM, DPS, and DAPS, with all methods evaluated on a single RTX5000 GPU. For the FFHQ dataset, SITCOM is at least twice as fast when compared to baselines. On ImageNet, SITCOM consistently requires much less run-time compared to DPS and DAPS. When compared to DDNM, SITCOM's run-time is similar or slightly lower. For example, on the super-resolution task, both SITCOM and DDNM average 1.34 minutes, but SITCOM achieves over a 2 dB improvement.

In Table 5, we report the average PSNR and LPIPS results using three more baselines: Denoising Diffusion Restoration Models (DDRM) (Kawar et al., 2022), Plug-and-Play (PnP) ADMM (Chan et al., 2016) (a non diffusion-based solver), and Regularization by Denoising with Diffusion (RED-Diff) (Mardani et al., 2023). The results of DDRM, PnP-ADMM, and RED-Diff are sourced from (Zhang et al., 2024). DDRM and PnP-ADMM present results for linear tasks whereas RED-Diff is used for the non-linear tasks. The results of SITCOM are as reported in Table 1.

When compared to DDRM and PnP-ADMM, SITCOM demonstrates notable improvements in both PSNR and LPIPS across all tasks and datasets. For instance, SITCOM achieves over a 5 dB improvement in random in-painting on both datasets. Compared to RED-Diff, SITCOM outperforms

| Task | Method | FFHQ | | | | ImageNet | | | |
|------|--------|------|------|------|------|------|------|------|------|
| | | PSNR (↑) | SSIM (↑) | LPIPS (↓) | Run-time (↓) | PSNR (↑) | SSIM (↑) | LPIPS (↓) | Run-time (↓) |
| Super Resolution 4× | DPS | $25.20_{\pm1.22}$ | $0.806_{\pm0.044}$ | $0.242_{\pm0.102}$ | $1.31_{\pm0.44}$ | $24.45_{\pm0.89}$ | $0.792_{\pm0.052}$ | $0.331_{\pm0.089}$ | $2.33_{\pm0.40}$ |
| | DAPS | $\underline{29.6}_{\pm0.67}$ | $\underline{0.871}_{\pm0.034}$ | $\mathbf{0.132}_{\pm0.088}$ | $1.24_{\pm0.43}$ | $\underline{25.98}_{\pm0.74}$ | $\underline{0.794}_{\pm0.09}$ | $\underline{0.234}_{\pm0.089}$ | $2.10_{\pm1.02}$ |
| | DDNM | $28.82_{\pm0.67}$ | $0.851_{\pm0.043}$ | $0.188_{\pm0.13}$ | $\underline{1.07}_{\pm0.35}$ | $24.67_{\pm0.78}$ | $0.771_{\pm0.06}$ | $0.432_{\pm0.34}$ | $\underline{1.38}_{\pm0.55}$ |
| | Ours | $\mathbf{30.95}_{\pm0.89}$ | $\mathbf{0.872}_{\pm0.045}$ | $\underline{0.137}_{\pm0.046}$ | $\mathbf{0.50}_{\pm0.34}$ | $\mathbf{26.89}_{\pm0.86}$ | $\mathbf{0.802}_{\pm0.057}$ | $\mathbf{0.224}_{\pm0.056}$ | $\mathbf{1.34}_{\pm0.45}$ |
| Box In-Painting | DPS | $23.56_{\pm0.78}$ | $0.762_{\pm0.034}$ | $0.191_{\pm0.087}$ | $1.52_{\pm0.43}$ | $20.22_{\pm0.67}$ | $0.69_{\pm0.034}$ | $0.297_{\pm0.077}$ | $1.55_{\pm0.44}$ |
| | DAPS | $24.41_{\pm0.67}$ | $\underline{0.791}_{\pm0.034}$ | $\underline{0.129}_{\pm0.067}$ | $1.33_{\pm0.42}$ | $21.79_{\pm0.34}$ | $0.734_{\pm0.045}$ | $\underline{0.214}_{\pm0.034}$ | $2.44_{\pm0.34}$ |
| | DDNM | $\underline{24.67}_{\pm0.067}$ | $0.788_{\pm0.024}$ | $0.229_{\pm0.055}$ | $\underline{1.02}_{\pm0.44}$ | $\underline{21.99}_{\pm0.54}$ | $\underline{0.737}_{\pm0.034}$ | $0.315_{\pm0.022}$ | $\underline{1.42}_{\pm0.45}$ |
| | Ours | $\mathbf{24.97}_{\pm0.55}$ | $\mathbf{0.804}_{\pm0.045}$ | $\mathbf{0.118}_{\pm0.022}$ | $\mathbf{0.37}_{\pm0.34}$ | $\mathbf{22.23}_{\pm0.44}$ | $\mathbf{0.745}_{\pm0.034}$ | $\mathbf{0.208}_{\pm0.023}$ | $\mathbf{1.23}_{\pm0.44}$ |
| Random In-Painting | DPS | $28.77_{\pm0.56}$ | $0.847_{\pm0.034}$ | $0.191_{\pm0.023}$ | $1.55_{\pm0.34}$ | $24.57_{\pm0.45}$ | $0.775_{\pm0.023}$ | $0.318_{\pm0.26}$ | $2.12_{\pm0.30}$ |
| | DAPS | $\underline{31.56}_{\pm0.45}$ | $\underline{0.905}_{\pm0.013}$ | $\underline{0.094}_{\pm0.012}$ | $1.42_{\pm0.45}$ | $28.86_{\pm0.67}$ | $0.877_{\pm0.021}$ | $0.131_{\pm0.044}$ | $2.01_{\pm0.34}$ |
| | DDNM | $30.56_{\pm0.56}$ | $0.902_{\pm0.013}$ | $0.116_{\pm0.023}$ | $\underline{1.25}_{\pm0.42}$ | $\underline{30.12}_{\pm0.45}$ | $\underline{0.917}_{\pm0.012}$ | $\underline{0.124}_{\pm0.032}$ | $\underline{1.89}_{\pm0.23}$ |
| | Ours | $\mathbf{33.02}_{\pm0.44}$ | $\mathbf{0.919}_{\pm0.012}$ | $\mathbf{0.0912}_{\pm0.013}$ | $\mathbf{0.47}_{\pm0.34}$ | $\mathbf{30.67}_{\pm0.45}$ | $\mathbf{0.918}_{\pm0.013}$ | $0.118_{\pm0.012}$ | $\mathbf{1.40}_{\pm0.34}$ |
| Gaussian Deblurring | DPS | $25.78_{\pm0.68}$ | $0.831_{\pm0.034}$ | $0.202_{\pm0.014}$ | $1.33_{\pm0.44}$ | $22.45_{\pm0.42}$ | $0.778_{\pm0.067}$ | $0.344_{\pm0.041}$ | $2.12_{\pm0.44}$ |
| | DAPS | $\underline{29.67}_{\pm0.45}$ | $\underline{0.889}_{\pm0.045}$ | $\underline{0.163}_{\pm0.033}$ | $2.15_{\pm0.37}$ | $26.34_{\pm0.55}$ | $0.836_{\pm0.034}$ | $\underline{0.244}_{\pm0.023}$ | $2.22_{\pm0.43}$ |
| | DDNM | $28.56_{\pm0.45}$ | $0.872_{\pm0.024}$ | $0.211_{\pm0.034}$ | $\underline{1.24}_{\pm0.34}$ | $\mathbf{28.44}_{\pm0.021}$ | $0.882_{\pm0.021}$ | $0.267_{\pm0.00}$ | $\underline{1.76}_{\pm0.33}$ |
| | Ours | $\mathbf{32.12}_{\pm0.34}$ | $\mathbf{0.913}_{\pm0.024}$ | $\mathbf{0.139}_{\pm0.045}$ | $\mathbf{0.45}_{\pm0.25}$ | $\underline{28.22}_{\pm0.45}$ | $\mathbf{0.891}_{\pm0.014}$ | $\mathbf{0.216}_{\pm0.021}$ | $\mathbf{1.34}_{\pm0.25}$ |
| Motion Deblurring | DPS | $23.78_{\pm0.78}$ | $0.742_{\pm0.042}$ | $0.265_{\pm0.024}$ | $1.65_{\pm0.34}$ | $22.33_{\pm0.727}$ | $0.726_{\pm0.034}$ | $0.352_{\pm0.00}$ | $2.21_{\pm0.40}$ |
| | DAPS | $\underline{30.78}_{\pm0.56}$ | $\underline{0.892}_{\pm0.034}$ | $\underline{0.146}_{\pm0.034}$ | $\underline{1.44}_{\pm0.34}$ | $\underline{28.24}_{\pm0.62}$ | $\underline{0.867}_{\pm0.024}$ | $\underline{0.191}_{\pm0.017}$ | $2.12_{\pm0.44}$ |
| | Ours | $\mathbf{32.34}_{\pm0.44}$ | $\mathbf{0.908}_{\pm0.028}$ | $\mathbf{0.135}_{\pm0.028}$ | $\mathbf{0.52}_{\pm0.34}$ | $\mathbf{29.12}_{\pm0.38}$ | $\mathbf{0.882}_{\pm0.025}$ | $\mathbf{0.182}_{\pm0.025}$ | $\mathbf{1.45}_{\pm0.31}$ |
| Phase Retrieval | DPS | $17.56_{\pm2.15}$ | $0.681_{\pm0.056}$ | $0.392_{\pm0.021}$ | $\underline{1.52}_{\pm0.42}$ | $16.77_{\pm1.78}$ | $0.651_{\pm0.076}$ | $0.442_{\pm0.037}$ | $\underline{2.18}_{\pm0.38}$ |
| | DAPS | $\underline{31.45}_{\pm2.78}$ | $\underline{0.909}_{\pm0.035}$ | $\underline{0.109}_{\pm0.044}$ | $1.85_{\pm0.32}$ | $\mathbf{26.12}_{\pm2.12}$ | $\underline{0.802}_{\pm0.023}$ | $\underline{0.247}_{\pm0.034}$ | $2.32_{\pm0.35}$ |
| | Ours | $\mathbf{31.88}_{\pm2.89}$ | $\mathbf{0.921}_{\pm0.067}$ | $\mathbf{0.102}_{\pm0.078}$ | $\mathbf{0.54}_{\pm0.45}$ | $\underline{25.76}_{\pm1.78}$ | $\mathbf{0.813}_{\pm0.032}$ | $\mathbf{0.238}_{\pm0.067}$ | $\mathbf{1.31}_{\pm0.45}$ |
| Non-Uniform Deblurring | DPS | $23.78_{\pm2.23}$ | $0.761_{\pm0.051}$ | $0.269_{\pm0.064}$ | $1.56_{\pm0.45}$ | $22.97_{\pm1.57}$ | $0.781_{\pm0.023}$ | $0.302_{\pm0.089}$ | $2.34_{\pm0.44}$ |
| | DAPS | $\underline{28.89}_{\pm1.67}$ | $\underline{0.845}_{\pm0.057}$ | $\underline{0.150}_{\pm0.056}$ | $\underline{1.41}_{\pm0.37}$ | $\underline{28.02}_{\pm1.15}$ | $\underline{0.831}_{\pm0.082}$ | $\underline{0.162}_{\pm0.034}$ | $\underline{2.23}_{\pm0.56}$ |
| | Ours | $\mathbf{31.09}_{\pm0.89}$ | $\mathbf{0.911}_{\pm0.056}$ | $\mathbf{0.132}_{\pm0.45}$ | $\mathbf{0.56}_{\pm0.37}$ | $\mathbf{29.56}_{\pm0.78}$ | $\mathbf{0.844}_{\pm0.045}$ | $\mathbf{0.147}_{\pm0.042}$ | $\mathbf{1.34}_{\pm0.44}$ |
| High Dynamic Range | DPS | $23.33_{\pm1.34}$ | $0.734_{\pm0.049}$ | $0.251_{\pm0.078}$ | $1.34_{\pm0.42}$ | $19.67_{\pm0.056}$ | $0.693_{\pm0.034}$ | $0.498_{\pm0.112}$ | $2.34_{\pm0.44}$ |
| | DAPS | $\underline{27.58}_{\pm0.829}$ | $\underline{0.828}_{\pm0.00}$ | $\underline{0.161}_{\pm0.067}$ | $\underline{1.26}_{\pm0.44}$ | $\underline{26.71}_{\pm0.088}$ | $\underline{0.802}_{\pm0.032}$ | $\underline{0.172}_{\pm0.066}$ | $\underline{2.12}_{\pm0.32}$ |
| | Ours | $\mathbf{28.52}_{\pm0.89}$ | $\mathbf{0.844}_{\pm0.045}$ | $\mathbf{0.148}_{\pm0.035}$ | $\mathbf{0.51}_{\pm0.42}$ | $\mathbf{27.56}_{\pm0.78}$ | $\mathbf{0.825}_{\pm0.037}$ | $\mathbf{0.162}_{\pm0.046}$ | $\mathbf{1.45}_{\pm0.41}$ |

Table 4: Average PSNR, SSIM, LPIPS, and run-time (minutes) of SITCOM and baselines using 100 test images from FFHQ and 100 test images from ImageNet with a **measurement noise level** of $\sigma_{\mathbf{y}} = 0.01$. The first five tasks are linear, while the last three tasks are non-linear (underlined). For each task and dataset combination, the best results are bolded, and the second-best results are underlined. Values after $\pm$ represent the standard deviation. All results were obtained using a **single RTX5000 GPU** machine. For phase retrieval, the run-time is reported for the best result out of four independent runs. This is applied for SITCOM and baselines.

| Task | Method | FFHQ | | ImageNet | |
|------|--------|------|------|------|------|
| | | PSNR (↑) | LPIPS (↓) | PSNR (↑) | LPIPS (↓) |
| Super Resolution 4× | DDRM (Kawar et al., 2022) | $\underline{27.65}$ | $\underline{0.210}$ | $\underline{25.21}$ | $\underline{0.284}$ |
| | PnP-ADMM (Chan et al., 2016) | 23.48 | 0.725 | 22.18 | 0.724 |
| | SITCOM (ours) | **30.68** | **0.142** | **26.35** | **0.232** |
| Box In-Painting | DDRM (Kawar et al., 2022) | $\underline{22.37}$ | $\underline{0.159}$ | $\underline{19.45}$ | $\underline{0.229}$ |
| | PnP-ADMM (Chan et al., 2016) | 13.39 | 0.775 | 12.61 | 0.702 |
| | SITCOM (ours) | **24.68** | **0.121** | **21.88** | **0.214** |
| Random In-Painting | DDRM (Kawar et al., 2022) | $\underline{25.75}$ | $\underline{0.218}$ | $\underline{23.23}$ | $\underline{0.325}$ |
| | PnP-ADMM (Chan et al., 2016) | 20.94 | 0.724 | 20.03 | 0.680 |
| | SITCOM (ours) | **32.05** | **0.095** | **29.60** | **0.127** |
| Gaussian Deblurring | DDRM (Kawar et al., 2022) | $\underline{23.36}$ | $\underline{0.236}$ | $\underline{23.86}$ | $\underline{0.341}$ |
| | PnP-ADMM (Chan et al., 2016) | 21.31 | 0.751 | 20.47 | 0.729 |
| | SITCOM (ours) | **30.25** | **0.235** | **27.40** | **0.236** |
| Motion Deblurring | PnP-ADMM (Chan et al., 2016) | $\underline{23.40}$ | $\underline{0.703}$ | $\underline{24.23}$ | $\underline{0.684}$ |
| | SITCOM (ours) | **30.34** | **0.148** | **28.65** | **0.189** |
| Phase Retrieval | RED-Diff (Mardani et al., 2023) | $\underline{15.60}$ | $\underline{0.596}$ | $\underline{14.98}$ | $\underline{0.536}$ |
| | SITCOM (ours) | **30.97** | **0.112** | **25.45** | **0.246** |
| Non-Uniform Deblurring | RED-Diff (Mardani et al., 2023) | **30.86** | $\underline{0.160}$ | **30.07** | $\underline{0.211}$ |
| | SITCOM (ours) | $\underline{30.12}$ | **0.145** | $\underline{28.78}$ | **0.160** |
| High Dynamic Range | RED-Diff (Mardani et al., 2023) | $\underline{22.16}$ | $\underline{0.258}$ | $\underline{22.03}$ | $\underline{0.274}$ |
| | SITCOM (ours) | **27.98** | **0.158** | **26.97** | **0.167** |

Table 5: Average PSNR and LPIPS results of our method and other baselines over 100 FFHQ and 100 ImageNet test images. The measurement noise setting is $\sigma_{\mathbf{y}} = 0.05$. The results of DDRM and PnP-ADMM (resp. RED-Diff) are sourced from Tables 1 and 3 (resp. 2 and 4) in (Zhang et al., 2024). The remaining results are as given in Table 1 of Section 4.

by 5 dB on FFHQ and more than 10 dB on ImageNet for phase retrieval. A similar trend is observed in the High Dynamic Range task. For non-linear non-uniform deblurring, although SITCOM performs better in terms of LPIPS, it reports approximately 1 dB (FFHQ) and 2 dB (ImageNet) less PSNR than RED-Diff, all without requiring external denoisers.

# F ABLATION STUDIES

## F.1 EFFECT OF THE NUMBER OF OPTIMIZATION STEPS $K$, & THE NUMBER OF SAMPLING STEPS $N$

In this subsection, we perform an ablation study on the number of optimization steps, $K$, and the number of sampling steps, $N$. Specifically, for the tasks of Super Resolution, Motion Deblurring, and Random In-painting, we run SITCOM using combinations from $N \in \{10, 20, 30\}$ and $K \in \{20, 30, 40\}$. The average PSNR results over 20 test images from the FFHQ dataset are presented in Table 6. As shown, for the first three tasks, SITCOM consistently achieves strong PSNR scores across all $(N, K)$ pairs, demonstrating that its performance is not very sensitive to variations in $(N, K)$ within these ranges as the results vary by nearly 1 dB. For High Dynamic Range tasks, we observe that the best results are obtained with $(N, K) = (20, 40)$. The selected $(N, K)$ values for our main results are listed in Table 8 of Appendix F.4.

| $(N, K)$ | (10, 20) | (10, 30) | (10, 40) | (20, 20) | (20, 30) | (20, 40) | (30, 20) | (30, 30) | (30, 40) |
|---|---|---|---|---|---|---|---|---|---|
| Super Resolution 4× | 29.654 | 29.771 | 29.815 | 29.913 | 29.952 | 29.961 | 30.009 | 30.027 | 30.033 |
| Motion Deblurring | 29.976 | 30.820 | 31.264 | 31.259 | 31.380 | 30.452 | 31.282 | 30.624 | 30.438 |
| Random Inpainting | 33.428 | 34.444 | 34.699 | 34.546 | 34.558 | 34.574 | 34.619 | 34.634 | 34.639 |
| High Dynamic Range | 25.902 | 26.290 | 27.873 | 26.957 | 27.104 | 27.874 | 27.171 | 27.127 | 26.806 |

Table 6: Effect of the number of sampling steps ($N$) and optimization steps per sampling iteration ($K$) on the tasks listed in the first column for SITCOM. The reported PSNR values are averaged over 20 FFHQ test images.

## F.2 EFFECT OF THE REGULARIZATION PARAMETER $\lambda$

In this subsection, we perform an ablation study to assess the impact of the regularization parameter, $\lambda$, in SITCOM. Table 7 shows the results across four tasks using various $\lambda$ values. Aside from phase retrieval, the effect of $\lambda$ is minimal. We hypothesize that initializing the optimization variable in Equation (S₁) with $\mathbf{x}_t$ is sufficient to enforce forward diffusion consistency in **C3**. Therefore, we set $\lambda = 1$ for phase retrieval and $\lambda = 0$ for the other tasks.

Additionally, for all tasks other than phase retrieval, we observed that when $\lambda = 0$, the restored images exhibit enhanced high-frequency details. For visual examples, see the results of $\lambda = 0$ versus $\lambda = 1$ in Figure 6.

| $\lambda$ | 0 | 0.05 | 0.5 | 1 | 1.5 |
|---|---|---|---|---|---|
| Super Resolution 4× | 29.952 | 29.968 | 29.464 | 29.550 | 29.288 |
| Motion Deblurring | 31.380 | 31.393 | 31.429 | 31.382 | 31.150 |
| Random Inpainting | 34.559 | 34.537 | 34.523 | 34.500 | 34.301 |
| Phase Retrieval | 31.678 | 31.892 | 32.221 | 32.342 | 32.124 |

Table 7: Ablation Study on the impact of the regularization parameter $\lambda$.

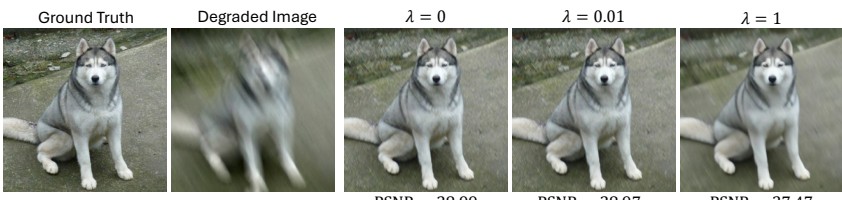

Figure 6: Results of running SITCOM using different regularization parameters in Equation (S₁) for the task of Motion deblurring.

## F.3 IMPACT OF THE STOPPING CRITERION FOR NOISY MEASUREMENTS

In this subsection, we demonstrate the impact of applying the stopping criterion in SITCOM when handling measurement noise. For the tasks of super resolution and motion deblurring, we run SIT-COM with and without the stopping criterion for the case of $\sigma_{\mathbf{y}} = 0.05$. The results are presented

in Figure 7. As shown, for both tasks, using the stopping criterion (i.e., $\delta > 0$) not only improves PSNR values compared to the case of $\delta = 0$, but also visually reduces additive noise in the restored images. This is because, without the stopping criterion , the measurement consistency enforced by the optimization in Equation ($S_1$) tends to fit the noise in the measurements.

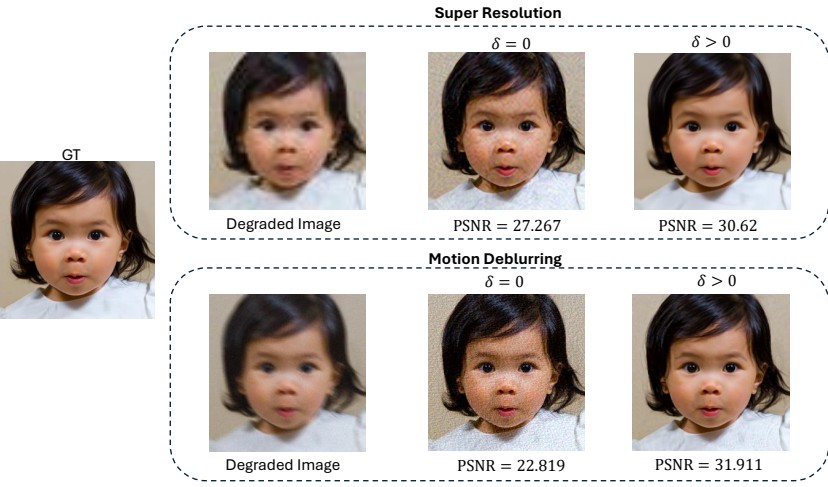

Figure 7: Impact of the stopping criterion in preventing noise overfitting. For the most right column, $\delta$ is set as in Table 8.

### F.4 COMPLETE LIST OF HYPER-PARAMETERS IN SITCOM

Table 8 summarizes the hyper-parameters used for each task in our experiments, as determined by the ablation studies in the previous subsections. Notably, the same set of hyper-parameters is applied to both the FFHQ and ImageNet datasets.

| Task | Sampling Steps $N$ | Optimization Steps $K$ | Regularization $\lambda$ | Stopping criterion $\delta$ for $\sigma_{\mathbf{y}} \in \{0.05, 0.01\}$ |
|---|---|---|---|---|
| Super Resolution $4\times$ | 20 | 20 | 0 | $\{0.051\sqrt{m_{\mathrm{SR}}}, 0.011\sqrt{m_{\mathrm{SR}}}\}$ |
| Box In-Painting | 20 | 20 | 0 | $\{0.051\sqrt{m}, 0.011\sqrt{m}\}$ |
| Random In-Painting | 20 | 30 | 0 | $\{0.051\sqrt{m}, 0.011\sqrt{m}\}$ |
| Gaussian Deblurring | 20 | 30 | 0 | $\{0.051\sqrt{m}, 0.011\sqrt{m}\}$ |
| Motion Deblurring | 20 | 30 | 0 | $\{0.051\sqrt{m}, 0.011\sqrt{m}\}$ |
| Phase Retrieval | 20 | 30 | 1 | $\{0.051\sqrt{m_{\mathrm{PR}}}, 0.011\sqrt{m_{\mathrm{PR}}}\}$ |
| Non-Uniform Deblurring | 20 | 30 | 0 | $\{0.051\sqrt{m}, 0.011\sqrt{m}\}$ |
| High Dynamic Range | 20 | 40 | 0 | $\{0.051\sqrt{m}, 0.011\sqrt{m}\}$ |

Table 8: Hyper-parameters of SITCOM for every task considered in this paper. The same set of hyper-parameters is used for FFHQ and ImageNet. The learning rate in Algorithm 1 is set to $\gamma = 0.01$ for all tasks, datasets, and measurement noise levels. For the stopping criterion column, $m_{\mathrm{SR}} = 64 \times 64 \times 3$, $m = 256 \times 256 \times 3$, and $m_{\mathrm{PR}} = 384 \times 384 \times 3$.

## G DETAILED IMPLEMENTATION OF TASKS AND BASELINES

The forward models of all tasks are adopted from DPS. We refer the reader to Appendix B of (Chung et al., 2023b) for details. For baselines, we used the codes provided by the authors of each paper: DPS[3], DDNM[4], DAPS[5], and DCDP[6]. Default configurations are used for each task.

---

[3] https://github.com/DPS2022/diffusion-posterior-sampling

[4] https://github.com/wyhuai/DDNM

[5] https://github.com/zhangbingliang2019/DAPS

[6] https://github.com/Morefre/Decoupled-Data-Consistency-with-Diffusion-Purification-for-Image-Restoration

# H QUALITATIVE RESULTS

Figure 8 presents results with SITCOM, DPS, and DAPS using ImageNet. See also Figure 9, Figure 10, Figure 11, and Figure 12 for more images.

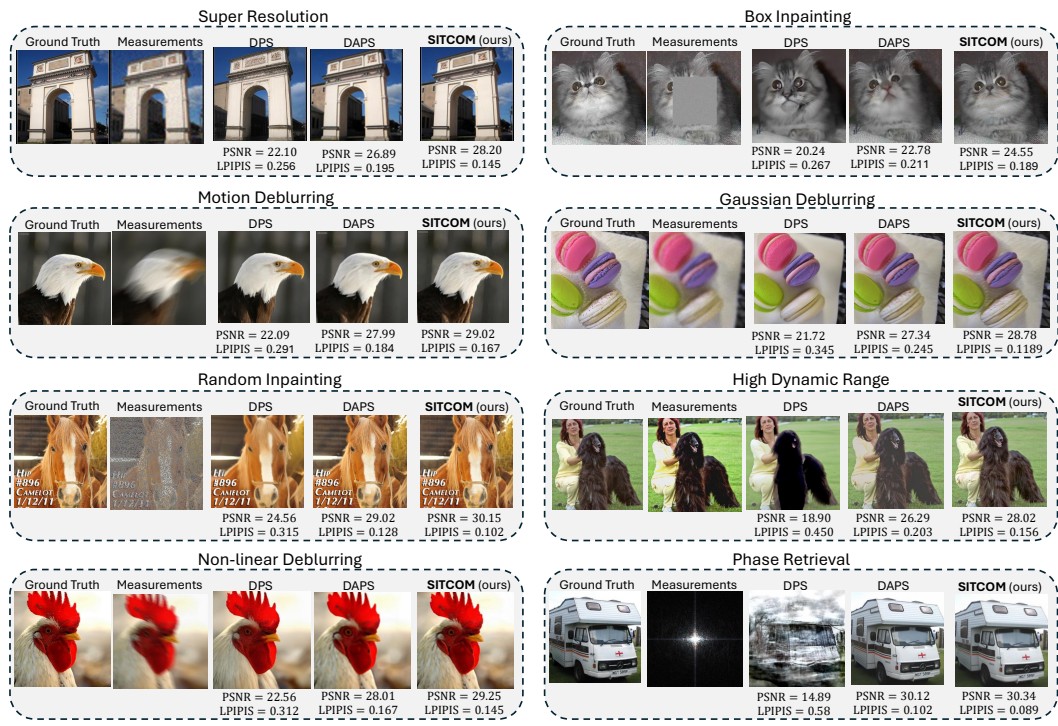

Figure 8: **Qualitative results on the ImageNet dataset** for five linear tasks and three non-linear tasks under measurement noise of $\sigma_{\mathbf{y}} = 0.05$. The PSNR and LPIPS values are given below each restored image.

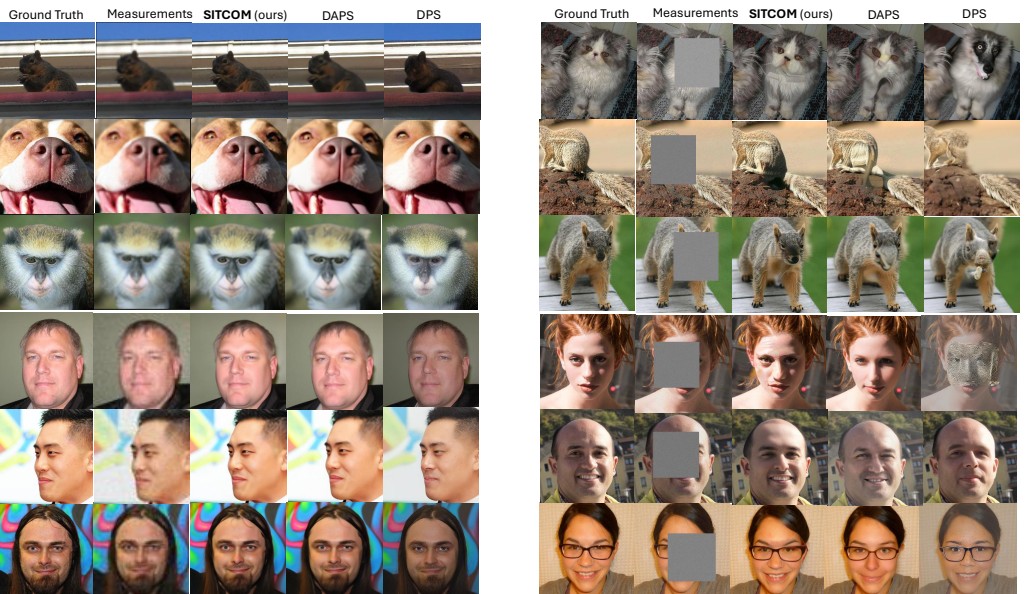

Figure 9: **Super resolution** (*left*) and **box inpainting** (*right*) results. First (resp. last) three rows are for the FFHQ (resp. ImageNet) dataset.

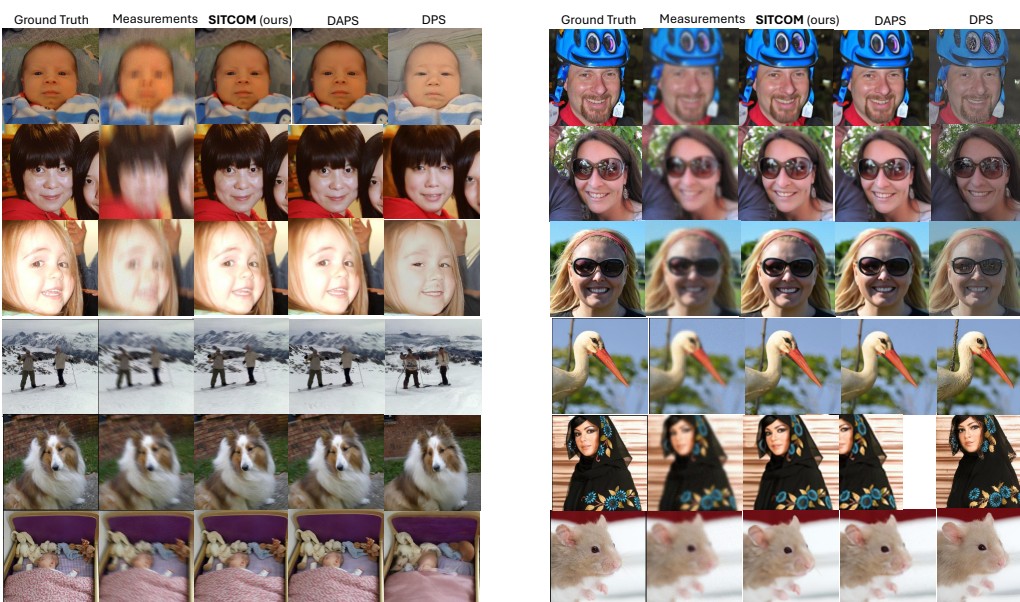

Figure 10: **Motion deblurring** (*left*) and **Gaussian deblurring** (*right*) results. First (resp. last) three rows are for the FFHQ (resp. ImageNet) dataset.

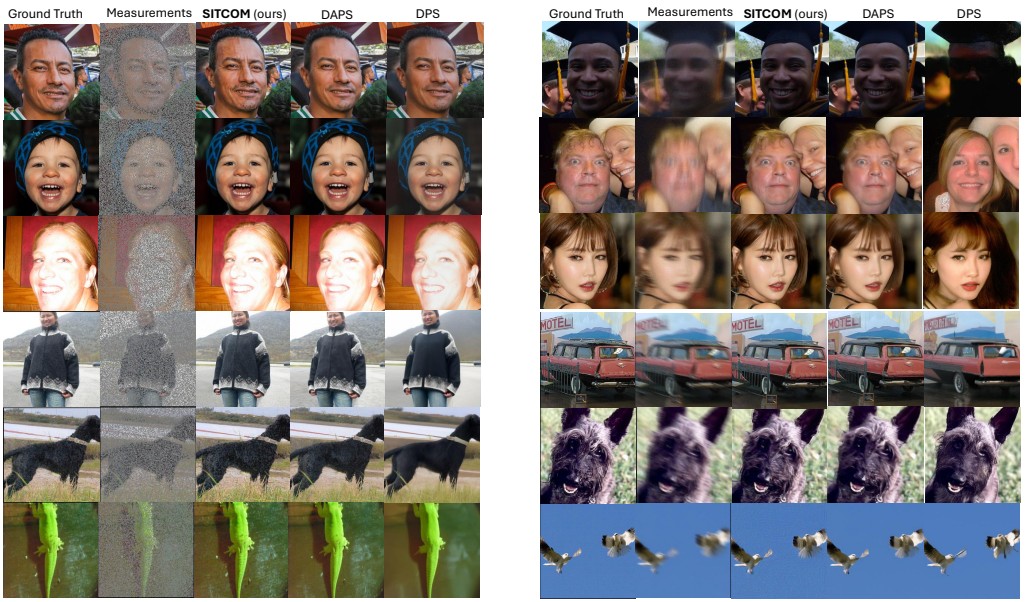

Figure 11: **Random inpainting** (*left*) and **non-linear (non-uniform) deblurring** (*right*) results. First (resp. last) three rows are for the FFHQ (resp. ImageNet) dataset.

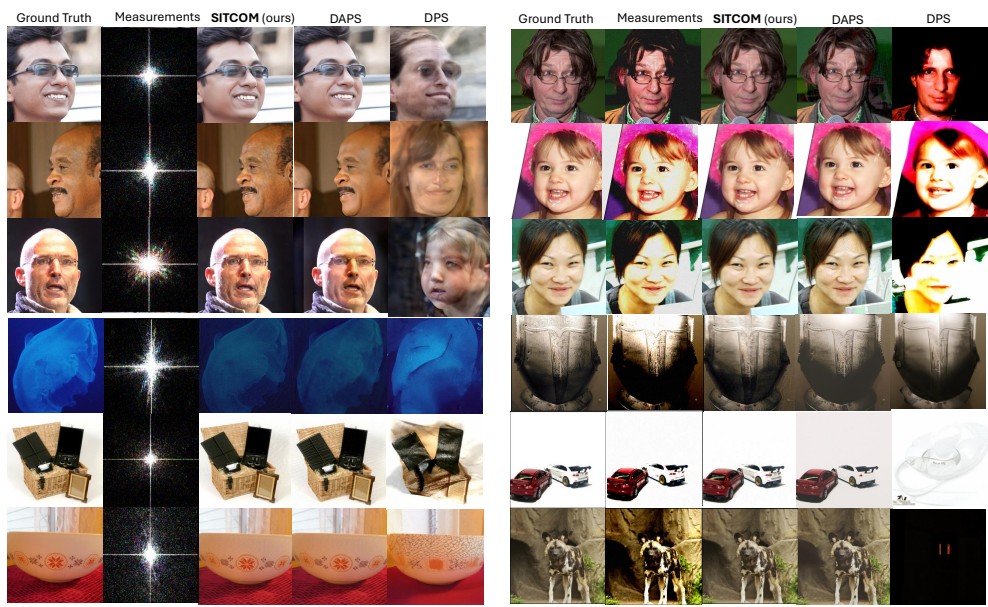

Figure 12: **Phase retrieval** (*left*) and **high dynamic range** (*right*) results. First (resp. last) three rows are for the FFHQ (resp. ImageNet) dataset.

