# OpenReview forum: "Step-wise Triple-Consistent Diffusion Sampling for Inverse Problems"
_ICLR.cc/2025/Conference — Submitted to ICLR 2025_

### Official Review · Reviewer_qJJQ · 2024-11-04

**Soundness:** 2
**Presentation:** 3
**Contribution:** 2
**Rating:** 6
**Confidence:** 4

**Summary:**

The paper introduces SITCOM (Step-wise Triple-Consistent Sampling), a new framework for solving inverse problems using diffusion models. SITCOM enforces three consistency conditions—measurement, backward, and forward consistency—to guide the sampling process and improve the fidelity of reconstructions. The approach is validated through comprehensive experiments on tasks such as deblurring, super resolution, and in-painting, with results showing competitive performance and reduced run-time compared to baseline methods.

**Strengths:**

nnovative Theoretical Contributions: The concept of backward consistency (Definition 1) is an original addition that, along with measurement and forward consistency, forms a robust framework for inverse problem solvers using diffusion models. This mathematical insight is well-formulated and adds depth to the field (lines 216–269).
Optimization-Based Sampling: The detailed explanation of the optimization problem in Equation (S1) and the associated algorithm (lines 324–431) shows a strong command of integrating theoretical principles with practical implementation.
Comprehensive Experiments: The paper includes results on various tasks and datasets, demonstrating the robustness of the proposed approach (Table 1).
Efficiency Improvements: The reported reduction in run-time, while maintaining or improving performance, is a notable practical advantage (lines 486–539).

**Weaknesses:**

Clarity in Backward Consistency: Definition 1 (lines 216–269) could benefit from more intuitive explanations or examples to make the concept clearer to a broader audience.
Absence of Mathematical Justifications: While the theoretical contributions are solid, the paper lacks formal mathematical justifications for key aspects:
Choice of Optimization Parameters: There is no in-depth analysis on the selection of K (number of optimization steps) or λ (regularization parameter). Providing this would strengthen the confidence in the method's robustness (lines 324–431).
Necessity of Triple Consistency: The paper argues for the simultaneous use of measurement, backward, and forward consistency but does not include formal proof or theoretical analysis justifying why all three are essential for the claimed performance (lines 216–377).
Convergence Analysis: A formal convergence analysis is missing, which could help clarify the reliability and scalability of the SITCOM sampler. While the algorithm shows empirical success, having mathematical insights into its convergence properties and error bounds would offer more confidence in its reliability and scalability.
Mixed Results in Some Tasks: The improvements in tasks like ImageNet Gaussian Deblurring and Phase Retrieval are less significant compared to other baselines (Table 1), which may raise questions about the trade-offs in complexity versus marginal performance gains.
Lack of Computational Details: Detailed metrics like Number of Function Evaluations (NFE) are not presented in the main manuscript (it is located at appendix F). Including these would provide a clearer view of the computational cost. Please consider reorganize the contents.
Comparison to Deterministic Samplers: The paper does not explore whether the SITCOM method could be adapted into a deterministic version like DDIM or address how it compares to the optimized noise schedules like EDM.

**Questions:**

1. Could the authors provide additional intuition or examples for Definition 1 to make backward consistency more accessible?
2. Are there any performance curves that illustrate the effect of varying the number of optimization steps K and sampling steps N on both image quality and computational efficiency?
3. Can the authors elaborate on how SITCOM performs when using fewer sampling or optimization steps and if there are trade-offs between quality and computational cost?
4. Is there potential for SITCOM to be adapted into a deterministic sampler like DDIM? If so, what modifications would be needed?
5. How does the method handle noise characteristics that deviate significantly from those tested in the paper?
6. Are there theoretical justifications for the choice of optimization parameters, or is there an analysis showing the method's convergence properties?
Overall: This paper makes a meaningful contribution by introducing a theoretically rich and empirically validated approach for inverse problem-solving using diffusion models. The addition of backward consistency is an interesting theoretical insight, and the experimental results are compelling. However, the paper could benefit from clearer explanations, formal justifications for certain methodological choices, and more detailed computational metrics. Addressing these points would enhance the completeness and accessibility of the work.

---

> ### Author Response · Authors · 2024-11-21
> **Reviewer qJJQ: Part 1/3**
>
> We would like to thank the reviewer for their comments.
>
> **[C1] Further clarifications/intuitions of the proposed backward consistency for a boarder audience.**
>
> We would like to thank the reviewer for this comment. We have incorporated a discussion on the motivation for the proposed backward consistency, drawing from the well-known Deep Image Prior (DIP) [A] regularization, on page 5. Additionally, we have revised Definition 1 to enhance clarity.
>
> In SITCOM, we apply the **network regularization during each sampling step such that we fully exploit the implicit regularization of the network to reduce the number of sampling steps and consequently the total running time**. We emphasize that the step-size backward consistency in SITCOM is the reason behind the reduced number of sampling steps.
>
> **[C2/Q2/Q6/Q3] (i) Formal mathematical justifications for the choices of $N$, $K$, and $\lambda$. (ii) Necessity of Triple Consistency. (iii) Convergence properties. (iv) Error bounds.**
>
> (i) We would like to thank the reviewer for their comment. Finding the optimal number of optimization steps and the optimal regularization parameter in SITCOM is challenging due to the presence of the neural network in the objective function in Eq.(S1). However, we conduct extensive results to shed more light into SITCOM's performance w.r.t. $N$, $K$, and $\lambda$:
>
> In Appendix~F.1, we report the results of pairs from $N\in \\{10,20,30\\}$ and $K\in \\{20, 30, 40\\}$. To fully address the reviewer's comment and infer if there is any trade-offs between quality and computational cost, we conduct **extended ablation studies** with $N \in \\{4,8,12,\dots,40\\}$ and $K\in \\{5, 10, 15,\dots,40\\}$. The results of GDB is given below (PSNR,run-time in seconds). We note that the results of the 8 tasks are given in (https://anonymous.4open.science/r/SITCOM-7539/Extended_Ablation.md).
>
> #### Gaussian Deblurring.
> | (N, K)   | K=5         | K=10        | K=15        | K=20        | K=25        | K=30        | K=35        | K=40        |
> |----------|-------------|-------------|-------------|-------------|-------------|-------------|-------------|-------------|
> | N=4      | 19.3,5.22   | 23.6,7.14   | 25.9,9.00   | 26.7,10.56  | 27.3,11.34  | 27.7,12.48  | 27.9,13.74  | 28.1,12.96  |
> | N=8      | 23.7,7.32   | 27.2,11.04  | 27.9,13.98  | 28.3,17.16  | 28.4,19.80  | 28.5,22.98  | 28.5,25.32  | 28.6,23.76  |
> | N=12     | 25.2,9.42   | 27.9,14.28  | 28.3,18.30  | 28.5,22.26  | 28.6,26.76  | 28.7,30.54  | 28.8,35.94  | 28.7,34.50  |
> | N=16     | 26.2,10.92  | 28.2,17.58  | 28.5,23.70  | 28.6,29.76  | 28.7,36.18  | 28.8,42.18  | 28.8,47.82  | 28.8,44.82  |
> | N=20     | 26.7,13.02  | 28.4,21.12  | 28.6,28.86  | 28.7,36.54  | 28.8,43.98  | 28.8,51.12  | 28.9,57.96  | 28.9,54.72  |
> | N=24     | 27.1,14.58  | 28.4,24.36  | 28.7,33.96  | 28.7,43.20  | 28.9,52.02  | 28.8,60.48  | 28.9,98.04  | 28.8,65.10  |
> | N=28     | 27.3,23.22  | 28.5,39.72  | 28.7,56.58  | 28.8,72.18  | 28.9,88.38  | 28.9,103.44 | 29.0,117.90 | 28.9,75.54  |
> | N=32     | 27.6,25.62  | 28.5,45.00  | 28.7,64.14  | 28.8,82.86  | 28.9,98.34  | 28.9,114.84 | 28.9,123.96 | 29.0,86.64  |
> | N=36     | 27.8,24.66  | 28.7,44.46  | 28.8,61.44  | 28.9,74.16  | 28.9,84.78  | 29.0,89.04  | 29.0,99.78  | 29.0,96.12  |
> | N=40     | 27.8,21.72  | 28.6,38.04  | 28.9,53.82  | 28.9,69.06  | 29.0,84.06  | 29.0,99.48  | 29.0,121.62 | 29.0,109.80 |
>
> We observe that with $N=12$ and $K=15$, we obtain 28.3dB (with only 18.3 seconds) whereas at $N=K=40$, we obtain 29 dB which required nearly 110 seconds. This indicates that there is indeed a trade-ff between computational cost and PSNR values. Furthermore, we observe that 61 entries (out of 80) achieve a PSNR values of more than or equal to 28 dB. This indicates that **SITCOM is not very sensitive to the choice** of $N$ and $K$. We note that we observe similar patterns for all tasks.
>
> [A] Deep Image Prior. CVPR18.

---

> ### Author Response · Authors · 2024-11-21
> **Reviewer qJJQ: Part 2/3**
>
> **[C2/Q2/Q6/Q3] (i) Formal mathematical justifications for the choices of $N$, $K$, and $\lambda$. (ii) Necessity of Triple Consistency. (iii) Convergence properties. (iv) Error bounds. [Continued]**
>
> (ii) For the necessity of the triple consistency, we refer the reviewer to studies in the second part of the global response.
>
> (iii) Due to the use of the neural network in our objective function, where the optimization variable is the input of the network, *analyzing the proposed steps theoretically is a very challenging task* including the convergence analysis of SITCOM. This is due the non-linearity of the network and the non-convexity of the loss landscape.
>
> In terms of the convergence properties of the optimization in Eq.(S1), we use the ADAM optimizer for at most $K$ steps. This means that if we look at the optimization at each sampling step, the theoretical convergence results of ADAM applies, such as the ones in [B].
>
> For the empirical convergence of SITCOM, we provide PSNR curves of SITCOM using different values of $N$ and $K$ to show the empirical convergence results: https://anonymous.4open.science/r/SITCOM-7539/convergence_plots.md.
>
> As observed, with $NK=400$, $(N,K)=(10,40)$ yields the best results. Notably, after completing the optimization steps at each sampling step, we observe a drop in PSNR, attributed to the application of the resampling formula in Eq.(S3).
>
> (iv)  We agree that deriving mathematical error bounds will benefit the understanding of the theoretical performance of SITCOM. However, this is particularly *challenging* when network regularization is involved. As noted in [C] in the context of unconditional diffusion: **the input error $\epsilon_{t+1}^{\text{cumu}}$ to the module $p_{\theta}(x_{t-1}|x_t)$ is parameterized by a non-linear network, it is not certain whether the input error is magnified or reduced in the error contained in its output $\epsilon_{t}^{\text{cumu}}$**. The added data consistency in SITCOM further complicates this analysis.
>
> To address this limitation, we added new numerical experiments and ablation studies as suggested to evaluate our method. Please see the second part of the global response.
>
> **[C3] (i) Improvements in ImageNet for PR and GDB tasks. (ii) NFEs.**
>
> We thank the reviewer for this comment/suggestion. As recommended, we will re-organize this part for further clarification.
>
> (i) We conjecture that the complexity is related to **the complexity of the task (ill-posedness) and the diversity of the dataset itself instead of our method**. ImageNet is a classification dataset with several labels that are unrelated semantically, whereas FFHQ is a dataset of mostly faces.
>
> For GDB, We are slightly lower than the SOTA in ImageNet. For PR, there is an interesting phenomena that still requires further investigation which we discuss in Appendix D. **In short, we observe that we have slightly more failure cases (i.e., very low PSNRs) than DAPS. However, once we don't fail, we obtain higher PSNRs**. To support this statement, we non-cherry pick a test image from ImageNet and run DAPS and SITCOM for 10 times. We report the PSNR for each. The results are given in (https://anonymous.4open.science/r/SITCOM-7539/PR_ImageNet_exp_vs_DAPS.md). The red results indicate failure cases. In SITCOM (resp. DAPS), we see 3/10 (resp. 2/10) cases. However, when we don't fail, our PSNRs are higher (the two bold results in SITCOM that is higher than the best results in DAPS).
>
> (ii) In terms of NFEs, given the proposed stopping criterion, **this results in at most $NK$ NFEs of the pre-trained model (forward pass), $NK$ backward passes through the pre-trained model, and $NK$ applications each for the forward operator and its adjoint to solve the optimization problem in Eq.(S1)**. With early stopping, the computational cost is lower. We discuss this in the paragraph before Section 3.6. In Tables 1 and 2 (Tables 1 and 4 in the revised paper), we opted to report the run-time instead of NFEs as different methods may apply the DM network differently and the computational cost could increase from other steps that don't use the DM network (for example, the optimization problem in DCDP).
>
> That said, we promise the reviewer to include NFEs in a subsequent version of our paper.
>
> [B] Convergence of Adam Under Relaxed Assumptions. NeuriPS23.
> [C] On Error Propagation of Diffusion Models. ICLR24.

---

> ### Author Response · Authors · 2024-11-21
> **Reviewer qJJQ: Part 3/3**
>
> **[C4/Q4] Deterministic Samplers for SITCOM such as DDIM and EDM.**
>
> This is a great question. We thank the reviewer for their suggestion. In SITCOM, we don't use the standard DDIM or DDPM sampling; instead, we only use the stochastic resample formula in Eq.(S3) to obtain $x_{t-1}$. *This is a stochastical sampler with randomness coming from the $\eta_t$ in Eq.(S3)*. We are not aware of a deterministic version of Eq.(S3).
>
> **However, your question has prompted us to recognize that the stochastic resampling formula can indeed be adapted into a deterministic form, akin to the modification from DDPM to DDIM.** We conducted an experiment where we compare SITCOM to the deterministic version of SITCOM for which we set $\eta_i$ deterministically via $\eta_i = x_N$ for all $i\in \\{N,\dots,1\\}$, where  $x_N \sim \mathcal{N}(0,I)$ is the initial input of the diffusion model. The following table includes the results of MDB with fixing $K=20$ (number of gradient updates per sampling step).
>
> |Method |  $\eta_i$ in Eq.(S3) | PSNR/run-time (seconds) |
> |---------|------|----------|
> | SITCOM ($N=20$)     | $\eta_i \sim \mathcal{N}(0,\mathbf{I})$    | 32.30/28.20 |
> | Deterministic SITCOM ($N=20$)  | $\eta_i = x_N$ for all $i$    |31.98/28.18 |
> | SITCOM ($N=10$)     | $\eta_i \sim \mathcal{N}(0,\mathbf{I})$    | 31.87/16.40 |
> | Deterministic SITCOM ($N=10$)  | $\eta_i = x_N$ for all $i$    |31.56/16.24 |
> | SITCOM ($N=5$)     | $\eta_i \sim \mathcal{N}(0,\mathbf{I})$    | 31.05/8.26 |
> | Deterministic SITCOM ($N=5$)  | $\eta_i = x_N$ for all $i$    |31.22/8.65 |
> | SITCOM ($N=4$)     | $\eta_i \sim \mathcal{N}(0,\mathbf{I})$    | 30.12/6.45 |
> | Deterministic SITCOM ($N=4$)  | $\eta_i = x_N$ for all $i$    |30.16/6.34 |
> | SITCOM ($N=3$)     | $\eta_i \sim \mathcal{N}(0,\mathbf{I})$    | 28.72/5.20 |
> | Deterministic SITCOM ($N=3$)  | $\eta_i = x_N$ for all $i$    |28.81/5.12 |
> | SITCOM ($N=2$)     | $\eta_i \sim \mathcal{N}(0,\mathbf{I})$    | 25.20/3.12 |
> | Deterministic SITCOM ($N=2$)  | $\eta_i = x_N$ for all $i$    |25.37/3.06 |
>
> We observe that at larger values of $N$, the stochasticity in Eq.($\texttt{S}_3$) returns better results. Interestingly, with $N\in \\{2,3,4,5\\}$, deterministic SITCOM returns slightly improved results. We will for sure include this discussion in a subsequent version of the paper.
>
> In the case of EDM, we think it is possible to extend SITCOM to EDM which would involve replacing the Tweedie’s formula in SITCOM with the second-order ODE solver from EDM. This requires investigating the best and most efficient way of incorporating the proposed backward consistency within steps of EDM.
>
> We thank the reviewer for this suggestion and promise to consider this direction for future investigation of our work.
>
> **[Q5] How does SITCOM handle noise characteristics that deviate significantly from those tested in the paper?**
>
> We thank the reviewer for this question. Here, we evaluate SITCOM with **additive measurement noise vector sampled from the Poisson distribution**. We use $N=K=20$ and $\lambda_{y} = 5$ (which determines the noise level as in DPS). We set $\delta$ (the stopping criterion) to different values (top row).
>
> Results for SR (linear IP) and NDB(non-linear IP) are given in the following table where we use 20 FFHQ test images with $m=256\times 256 \times 3$. The first row shows the values of $a$ for which $\delta = a\sqrt{m}$.
>
> |Task | $0$ |  $0.02$ | $0.035$ |$0.05$ |$0.06$ |$0.061$ |$0.065$ | $0.08$ |$0.1$ |$0.5$ |
> |---------|------|----------|-|-|-|-|-|-|-|-|
> | Super-Resolution  | 24.52/47.30 | 27.80/40.19    | 29.19/38.11 |30.42/36.23    | 30.49/28.22 |30.43/28.21    | 30.37/24.16 | 29.67/24.32 | 26.77/22.08    | 25.12/15.88 |
> | Non-uniform  Deblurring | 25.04/43.42 | 28.60/40.34    | 30.17/37.52 | 30.22/35.44    | 30.24/32.02 |30.23/29.31    | 30.15/27.51 |29.71/24.02|29.03/22.25    | 26.44/15.15 |
>
>
> As observed, the results of setting $\delta$ to values between $0.035\sqrt{m}$ to $0.08\sqrt{m}$ return PSNR values within approximately 1 dB. **This indicates that SITCOM can perform reasonably well on noise sources other than Gaussian with different values of the stopping criterion (even if it is set for the additive Gaussian measurement noise case)**. We will consider more noise settings in a subsequent version of the paper.

---

> > ### Author Response · Authors · 2024-11-25
> > **A friendly and gentle reminder**
> >
> > We would like to express our sincere gratitude to the reviewer once again for their insightful comments.
> >
> > As the open discussion period is drawing to a close, we would be deeply grateful if the reviewer could kindly respond to our rebuttal. This would provide us with the opportunity to address or clarify any remaining concerns thoroughly.

---

> > > ### Comment · Reviewer_qJJQ · 2024-12-02
> > >
> > > I really appreciate the effort of the authors to clarify few questions and provide more evidences to support their paper. Although the quality of the paper has been improved, it is not to sufficient to boost the score from 6 to 7.

---

### Official Review · Reviewer_iQjf · 2024-11-04

**Soundness:** 1
**Presentation:** 3
**Contribution:** 2
**Rating:** 3
**Confidence:** 5

**Summary:**

This paper identifies the key limitations in previous DMs for IPs and summaries, referred to as the triple-consistency conditions. The newly designed optimization-based method, SITCOM, achieves comparable or even superior restoration results with significantly reduced computational time. The paper demonstrates the superiority of the proposed method across commonly used benchmarks, including five linear tasks and three nonlinear tasks.

**Strengths:**

1. The intuitive explanation of the triple-consistency conditions is clear and reasonable. The paper identifies the potential factors that prevent existing methods from balancing quality and efficiency, summarizing as the triple-consistency conditions.

2. The paper consistently achieves comparable or superior empirical results across commonly used linear and nonlinear tasks as shown in Table 1.

**Weaknesses:**

1. **Incremental Modification and Unfair Comparisons**
   The major issue with the paper is that it presents only an incremental modification over existing work and makes unfair comparisons.

   1. *Theoretical and Motivational Perspective*
      Although the paper is the first to propose and formulate the triple-consistency conditions formally, each has already been individually adopted and utilized in other works. Specifically, measurement consistency is used in [4, 5, 6, 7], backward consistency in [1, 2, 3], and forward consistency in [4, 7]. The paper provides only intuitive explanations for each condition in Sections 3.1, 3.2, and 3.3, lacking clear evidence of how each contributes to the final restoration performance. A straightforward ablation study—removing each consistency condition from the proposed algorithm, SITCOM, and comparing performance with the full SITCOM while keeping other parameters—would give clearer hints of the significance of each condition. Unfortunately, the experiments fail to control variables, causing the conclusions of the paper potentially misleading.

   2. *Technical Perspective*
      The proposed method, SITCOM (Algorithm 1 in the paper), differs from DCDP only in two aspects:
      * (1) Changing the optimization variable from $\hat{x}_0$ to $\hat{v}_i$, where these variables are connected by the Tweedie-network denoiser $\hat{x}_0' = f(\hat{v}_i; t, \theta)$.
      * (2) Adding an early stopping criterion (line 5 in Algorithm 1).

      My main concern is with (1). The paper claims this change is introduced to achieve backward consistency but comes at the cost of computing backward gradients through the Tweedie-network denoiser $f$. Consequently, when keeping other parameters the same (e.g., diffusion steps $N$, number of gradient updates $K$, stopping criterion $\delta$), SITCOM is significantly slower than DCDP due to the additional backward pass through $f$. However, the paper compares SITCOM with smaller $N$ and $K$ to DCDP with much larger $N$ and $N$ and concludes that SITCOM is more efficient in achieving comparable results. I strongly suggest that the authors conduct a detailed and fair comparison between SITCOM and DCDP by fixing all other parameters and only changing the optimization variables. Otherwise, it is unclear whether the observed efficiency gains are due to backward consistency or simply fewer diffusion and gradient steps.

2. **Other Minor Issues**
   1. The paper lacks comparison and discussion of several recently proposed methods [5, 6, 8].
   2. There is no formulation or experimentation on solving inverse problems with latent diffusion models, which are included in the baseline papers used [4, 7].
3. **Non-anonymous issue.** Files in the given code link [https://anonymous.4open.science/r/SITCOM-7539/README.md](https://anonymous.4open.science/r/SITCOM-7539/README.md) (`SITCOM.py`, `SITCOM_with_noise.py` and `SITCOM_with_noise_imagenet.py`, etc.)  contain non-anonymous information.

**References**

[1] Chung, Hyungjin, et al. "Diffusion Posterior Sampling for General Noisy Inverse Problems." *ICLR 2023*.

[2] Song, Jiaming, et al. "Pseudoinverse-Guided Diffusion Models for Inverse Problems." *ICLR 2023*.

[3] Song, Jiaming, et al. "Loss-Guided Diffusion Models for Plug-and-Play Controllable Generation." *ICML 2023*.

[4] Li, Xiang, et al. "Decoupled Data Consistency with Diffusion Purification for Image Restoration." arXiv preprint arXiv:2403.06054 (2024).

[5] Wu, Zihui, et al. "Principled Probabilistic Imaging using Diffusion Models as Plug-and-Play Priors." arXiv preprint arXiv:2405.18782 (2024).

[6] Xu, Xingyu, et al. "Provably Robust Score-Based Diffusion Posterior Sampling for Plug-and-Play Image Reconstruction." arXiv preprint arXiv:2403.17042 (2024).

[7] Zhang, Bingliang, et al. "Improving Diffusion Inverse Problem Solving with Decoupled Noise Annealing." arXiv preprint arXiv:2407.01521 (2024).

[8] Dou, Zehao, et al. "Diffusion Posterior Sampling for Linear Inverse Problem Solving: A Filtering Perspective." *ICLR 2024*.

**Questions:**

Please address my concerns in the Weaknesses part.

**Details Of Ethics Concerns:**

No Ethics Concerns

---

> ### Author Response · Authors · 2024-11-21
> **Reviewer iQjf: Part 1/3**
>
> **[C1] (i) How the proposed consistencies relate to other works. (ii) Validity of the comparisons. (iii) Suggested study to show the impact of each consistency in SITCOM.**
>
> We thank the reviewer for their comment.
>
> **(i) How the proposed consistencies relate to other works**:
> We agree with the reviewer that *measurement consistency*, in some form, is necessary for (and used in) all IP solvers. The *forward consistency*, as described/stated in Section 3.1 (Issue 2 and the paragraph before Eq.(8)), is adopted from (Lugmayr et al., 2022) based on their work's observation/intuition.
>
> The technical novelty of SITCOM can be highlighted as:
> - **Formulating the three consistencies**: We present a systematic formulation of the conditions where the goal is to improve the accuracy of each intermediate reconstruction, thereby reducing the number of sampling steps required.
> - **Introducing the step-wise Backward consistency**: We introduce step-wise network regularization for the backward consistency by leveraging the implicit bias of the network at each iteration. In this rebuttal, we present new numerical results demonstrating the benefits of enforcing the proposed backward consistency (see the third experiment of the second part of the global response).
> - **Integrating all consistencies:** The formulation, initialization, and regularization of the optimization problem in Eq.(S1) along with the preceding two steps ensure that all consistencies are enforced without violation. Note that the initialization (and regularization) in Eq.(S1) is partially used to ensure closeness such that the forward consistency is satisfied.
>
> It is worth noting that while some previous works, such as RED-diff and DMPlug, also utilize the implicit bias of the network, **they adopt the full diffusion process as a regularizer, applied only once**. In contrast, our method uses the neural network as the regularizer at each iteration and focuses specifically on reducing the number of sampling steps for a given level of accuracy. We compared our approach with these methods to highlight this advantage (see the first part of the global response). We have also further clarified this novel aspect. Please refer to the highlighted changes on pages 5 and 6 of the revised manuscript.
>
> In addition to the empirical comparisons with DPS [1], we refer the reviewer to the first study in our global response to highlight the impact of the iterative application of measurement consistency.
>
> In what follows, we discuss the differences between SITCOM and two other papers.
>
> For PGDM [2], the authors assumes that $p(x_0|x_t)$ follows a Gaussian distribution which is then used to approximate the intractable term $\nabla_x \log p_t(y|x_t)$. This approximation is used to construct a guidance vector (with the measurement and forward operator) used in the remapping. To this end, there are several differences between SITCOM and PGDM.
> - We use forward consistency (i.e., Eq.(S3)) for the remapping whereas PGDM uses a DDIM step to obtain $x_{t-1}$ (i.e., Eq.(7) with guidance in their paper). We refer the reviewer to the second study in the global response to show the impact of the resampling formula in SITCOM.
> - Measurement consistency in SITCOM is applied through an optimization problem that also integrates backward consistency given Definition 1, whereas PGDM integrates the measurements vector into the guidance vector.
> - In SITCOM, the input to the Tweedie's formula is our optimization variable in (i.e., Eq.(S3)) whereas in PGDM, Tweedie's is used to obtain an estimate (without measurement consistency) that will later be used to construct the guidance vector for the remapping. See Algorithm 1 in Appendix A of PGDM. F
>
> or LGD [3], the authors used the same assumption in PGDM with the difference of using Monte Carlo sampling. The authors proposed to use the average of multiple samples to approximate the intractable term. In general, all these methods do not use a step-wise network regularization for the backward consistency.
>
> **(ii) Validity of the comparisons**:
> In terms of comparisons, we are using the **same tasks, test images, pre-trained models, and measurement noise levels**. If the reviewer is referring to the hyper-parameters in the baselines, we would like to clarify that we are using the configuration set by the authors in their codes and recommended in their papers. Additionally, we only use tasks evaluated by the authors of the papers. For example, we don't include phase retrieval results for DMPlug as this task was not considered. *It would be great if the reviewer could elaborate more on this point.*
>
> **(iii) Suggested study to show the impact of each consistency in SITCOM.**:
> We thank the reviewer for the suggested experiment about highlighting the impact of every step in SITCOM. Please see the second part our global response.

---

> ### Author Response · Authors · 2024-11-21
> **Reviewer iQjf: Part 2/3**
>
> **[C2] (i) Technical Differences between DCDP and SITCOM. (ii) Comparing SITCOM and DCDP with the same $N$ and $K$.** We would like to thank the reviewer for their comment/suggestion.
>
> (i) From a higher level perspective, the motivation of SITCOM is different from DCDP. In DCDP, the DM is used as a **diffusion purifier** [A], where the forward and reverse processes are used at inference. As the DM is trained on clean images, the intuition in DCDP is that when the reverse sampling steps (without measurement consistency) are applied on a noisy version of some input image, the DM-purified one will be closer to the learned/clean image distribution (see Theorem 3.1 in [A]). Then, the measurement consistency is applied like in Eq.(5) on the DM-purified version (denoted as $v_k$ in Algorithm 1 in DCDP). In SITCOM, the motivation is to find an efficient modification of the reverse sampling steps such that we can achieve accelerated reconstructions by proposing a new method for enforcing the backward consistency through the proposed network regularization.
>
> The **formulation/encoding** of the **DM network** is also significantly different as the output of the DM in DCDP is only used as initialization for their optimization problem (the data fidelity function in the DCDP algorithm). In SITCOM, the optimization variable is the input of the *DM network* that is formulated to enforce measurement- and backward-consistency (through the proposed step-wise network regularization).
>
> In SITCOM, **the gradients need to be computed w.r.t. the input of the network, not parameters**. The reviewer is correct, the computation of these gradients requires a backward pass through the network parameters which is not needed in DCDP. However, the proposed backward consistency allowed us to reduce the sampling and optimization steps and still achieve leading or competitive PSNRs as we discuss next.
>
> (ii) The authors in DCDP propose *two versions* of DCDP in the pixel space. Version 1 simply uses the SDE steps to apply purification (using a linear decaying schedule), whereas Version 2 uses Tweedies formula to approximate this step (the $\texttt{DPUR}$ function in DCDP). In our experiments, we only consider version 1 based on the authors' observation that version 2 results in lower perceptual quality because of the "regression to the mean" effects. To handle measurement noise, DCDP modifies the purification schedule, $T_k$ (per sampling step $k$ ), such that the decay does not go to $T_k=0$, whereas in SITCOM, we use a stopping criterion.
>
> Following the reviewer's suggestion, we compare SITCOM with DCDP using the same $N$ and $K$, denoted as $K$ and $\tau$ in DCDP. We set $\sigma_y = 0$ because SITCOM and DCDP handles measurement noise differently.
>
>  NDB:
>  |Method | $N$ | $K$ |  PSNR | run-time |
> |---------|------|----------|-|-|
> | SITCOM    | 20 | 20 | 31.24    | 29.02 |
> | DCDP    | 20 | 20 | 21.10    | 18.57 |
> | SITCOM     | 10    | 30 | 29.2 | 20.120 |
> | DCDP   | 10    | 30 | 20.60    | 12.00 |
> | SITCOM  | 30    | 10 |   28.88    | 20.08 |
> | DCDP   | 30    | 30  | 21.26    | 24.02 |
> | DCDP   | 40    | 40  | 22.40    | 32.50 |
> | DCDP   | 50    | 50  | 25.32    | 60.20 |
> | DCDP   | 60    | 60  | 27.20    | 81.40 |
> | DCDP   | 80    | 80  | 27.60    | 140 |
> | DCDP   | 90    | 90  | 28.67    | 178 |
> | DCDP   | 100    | 100  | 29.20    | 225 |
> | DCDP   | 150    | 100  | 29.70    | 281 |
> | DCDP   | 175    | 100  | 30.10    | 312 |
> | DCDP (default for NDB)   | 200 | 100 | 30.42    | 344 |
>
> For non-linear deblurring, DCDP requires a large number of $N$ and $K$ to achieve comparable PSNRs (the last row) which results in extended run-times.
>
> GDB:
> |Method | $N$ | $K$ |  PSNR | run-time |
> |---------|------|----------|-|-|
> | SITCOM    | 10 | 10 | 31.6    | 8.72 |
> | DCDP   | 10 | 10 | 29.52    | 10.50 |
> | SITCOM    | 20 | 20 | 32.28    | 28.12 |
> | DCDP   | 20 | 20 | 31.67    | 16.24 |
> | SITCOM     | 10    | 30 | 31.66 | 22.11 |
> | DCDP   | 10    | 30 | 31.11    | 12.42 |
> | SITCOM  | 30    | 10 |   31.34    | 22.08 |
> | DCDP   | 30    | 10  | 30.81    | 12.04 |
> | SITCOM  | 50    | 10 |   32.32    | 32.22 |
> | DCDP   | 50    | 10  | 31.69    | 26.20 |
> | SITCOM  | 100    | 10 |   32.34    | 51.23 |
> | DCDP   | 100    | 10  | 31.71    | 42.45 |
> | SITCOM  | 200    | 10 |   32.41    | 124.20 |
> | DCDP   | 200    | 10  | 31.76    | 112.10 |
> | SITCOM  | 500    | 10 |   32.46    | 320.20 |
> | DCDP   | 500    | 10  | 31.82    | 309.10 |
>
> The reviewer is correct. SITCOM is slower than DCDP when the number of iterations is fixed due to the need to backpropagate. However, SITCOM allows to reduce $N$ and $K$ while achieving competitive PSNRs. We can see from the table that the PSNR of SITCOM with $N=10$ matches DCDP with 20, therefore we claimed that SITCOM reduced the total run time.
>
> **For any fixed $N$ and $K$, on both tables, SITCOM always returns higher PSNR**.
>
> [A] Diffusion Models for Adversarial Purification. ICML22.

---

> ### Author Response · Authors · 2024-11-21
> **Reviewer iQjf: Part 3/3**
>
> **[C3] Minor: Comparison with [5,6,8].**
>
> Thank you for your comment. For DMPlug [5] and FPS [8], we include comparison results in the global response. For DPnP [6], the results are very similar to DPS and due to the time limit of the rebuttal, we opted not to compare with them for now. However, we promise to include them in a subsequent version of the paper.
>
> **[C4] Code link.**
>
> We would like to thank the reviewer for pointing this out. Once the reviews were released, we immediately fixed this issue.
>
> **[C5] Minor: Formulation of SITCOM in the latent space and comparison results.**
>
> We thank the reviewer for their comment. Latent-DMs were originally introduced to speed up the sampling procedure as the forward and reverse processes are applied in a lower dimensional space. This was the motivation of methods like ReSample and PSLD. **However, in lower dimensional spaces, it is required to have two pre-trained models: The pre-trained DM and the pre-trained encoder and decoder, used to convert between the pixel and latent spaces**. For solving IPs with latent DMs, a **backward pass through the decoder** (and possible forward passes through the encoder as is done in Latent-DCDP) is needed, which is an **additional** computational cost that may not result in a much reduced computational time (see the run-time results in Table 10 of ReSample). This is demonstrated in ReSample where the measurement consistency optimization problem was only applied for sampling time indices in set $\mathcal{C}$ of their algorithm. This is the main reason we opted not to consider SITCOM with latent space DMs.
>
> While the formulation for extending SITCOM to the latent space *may seem straightforward* (as we can simply apply the decoder to the argument of $\mathcal{A}(\cdot)$ in Eq.(S1)), we still need to verify it and check whether other techniques (such as the ReEncode function in latent-DCDP) are needed.
>
> We promise to further investigate this either in a subsequent version of the paper or for future work.
>
> To fully address the reviewer's comment, in the following table, we include results of pixel-space SITCOM and latent space methods.
>
> **Dataset**: FFHQ. **Metric**: PSNR. **Noise level**: 0.05. **Source**: Table 1 and Table 2 of DAPS.
>
> |Method |  SR | BIP | RIP | GDB |
> |---------|------|----------|-|-|
> | Latent-DAPS     | 27.48    |  23.99 | 27.93 | 27.00 |
> | PSLD            | 24.35    | 24.22  | 23.27 | 22.31 |
> | ReSample        | 23.29    | 20.6   | 26.39 | 27.41 |
> | SITCOM (Ours)   | **30.68**|  **24.68** | **32.05** | **30.25** |
>
> It is important to note that latent-DM IP methods **are not fully comparable to pixel space SITCOM as different pre-trained DMs were used**. However, similar to ReSample (where they compared their latent-DM IP solver with pixel-space ones), we included the above results.

---

> > ### Author Response · Authors · 2024-11-25
> > **A friendly and gentle reminder**
> >
> > We would like to express our sincere gratitude to the reviewer once again for their insightful comments.
> >
> > As the open discussion period is drawing to a close, we would be deeply grateful if the reviewer could kindly respond to our rebuttal. This would provide us with the opportunity to address or clarify any remaining concerns thoroughly.

---

> > > ### Comment · Reviewer_iQjf · 2024-11-27
> > > **Official Comment by Reviewer iQjf**
> > >
> > > I sincerely appreciate the author’s hard work on the rebuttal. Providing such an extensive set of additional results within the limited timeframe was a significant effort. However, after carefully reviewing the provided materials, I still have several main concerns, which I have summarized below:
> > >
> > > 1. **Global Response: Part 2/2, Sec 3: Impact of the Backward Consistency**. The reported numbers in this section raise some concerns. If I understand correctly, replacing backward consistency in SITCOM [by substituting Eq. (S1) and Eq. (S2) with Eq. (5)] should essentially make it equivalent to DCDP, which relies on forward and data consistency. Can you clarify why the results in this case are significantly worse than those reported for DCDP? Additionally, the authors mentioned that these comparisons were conducted under the "same runtime," as determined by the number of gradient updates. However, SITCOM with backward consistency requires backpropagation through the network for optimization, which is computationally more expensive. This is especially true for tasks like SR and NDB, where the gradient of forward functions is relatively efficient to compute. Given this, how was the runtime equivalence established, and how does it affect the reported performance?
> > >
> > > 2. **[C2] (ii)**: Again, I completely understand that providing such extensive results in a short time is challenging, and I sincerely appreciate the effort put into addressing my concerns. However, I still doubt the validity of some of the reported numbers. To illustrate, I’ve retrieved part of the table for reference:
> > >
> > >    | Method | $N$  | $K$  | Runtime |
> > >    | ------ | ---- | ---- | ------- |
> > >    | DCDP   | 10   | 30   | 12.00   |
> > >    | DCDP   | 30   | 30   | 12.02   |
> > >    | DCDP   | 60   | 60   | 81.40   |
> > >    | DCDP   | 90   | 90   | 178     |
> > >
> > >  The numbers in this table seem inconsistent, and I suspect there may be a typo or something. These discrepancies raise questions about the reliability of the reported results and whether similar mistakes might exist elsewhere in the additional results. I strongly believe that these results need careful check. A new round of peer review would provide the necessary time to thoroughly verify the validity and ensure the work's overall quality.
> > >
> > > **Additional Minor Comments:**
> > >
> > > 1. **Global Response: Part 1/2**. I appreciate the additional comparisons provided by the authors. However, the presentation of the results appears quite messy and difficult to interpret. It would be much more straightforward if the authors could provide a benchmarking comparison of the methods under consistent settings and across a unified set of tasks.
> > > 2. **Global Response: Part 2/2, Sec 1: Impact of the iterative application of measurement consistency in the first step of SITCOM**. The results for SR and NDB are reasonable and align with my expectations because vanilla regularized optimizations for these tasks result in blurry reconstructions with high PSNR but poor LPIPS scores. That said, I would encourage the authors to include additional nonlinear tasks, such as phase retrieval, and report both PSNR and LPIPS metrics to provide a more comprehensive evaluation.
> > >
> > >
> > >
> > > **Summary:**
> > >
> > > I will maintain my original score of **3: Reject**. The primary reasons:
> > >
> > > 1. **Messy Presentation of Additional Results**. The additional results provided during the rebuttal phase are disorganized and challenging to interpret. These results should have been incorporated into the original paper in a more organized and coherent way to make the storyline complete and convincing. A new round of peer review would be beneficial.
> > > 2. **Insufficient Analysis of the Proposed Method**. The original paper lacks significant and convincing analysis regarding the effectiveness of the three consistencies. This omission makes it difficult to assess the true impact and efficacy of the proposed method. And the additional results don't fully address my initial concerns.

---

> > > > ### Author Response · Authors · 2024-11-27
> > > > **Thank you for your response. Part 1/2**
> > > >
> > > > We truly thank the reviewer for their detailed comments/questions and their engagement in the discussion. We are glad that the reviewer acknowledges our efforts to include the additional results in the rebuttal. Please see our response below.
> > > >
> > > > ## Resemblance between SITCOM without backward consistency and DCDP. Clarification of why the results in this case are significantly worse than those reported for DCDP.
> > > >
> > > > The reviewer is correct that **SITCOM without backward consistency** shares resemblance with **DCDP version 2**. However, there is a major difference: The resampling formula (i.e., Eq.(S3)) in DCDP is used for diffusion purification, not for remapping as is done in SITCOM. In SITCOM, it is employed to remap the image back to $t-1$, whereas in DCDP version 2, it is used to add noise as part of the forward process of diffusion purification. In the latter, DCDP uses $T_k$ instead of $t-1$ where $T_k$ is an adjustable parameter. The authors of DCDP adopts a linear decaying schedule for $T_k$ that differs from task to task (refer to the third bullet point in Appendix A1 of DCDP).
> > > >
> > > > **The results we report in the above DCDP tables are for DCDP version 1 (not version 2)** where the SDE steps (without data consistency) are used instead of Tweedie's formula for diffusion purification. As stated in our response above, we used DCDP version 1 as the authors stated that the performance of version 2 is of low perceptual quality.
> > > >
> > > > We note that SITCOM without backward consistency is more similar to ReSample, but in the pixel space.
> > > >
> > > > It is important to note that *the results in the impact of the backward consistency experiment are for $t'>0$, not at $t=0$*. **The goal here is to show intermediate reconstructions under the same number of gradient updates. This setting is used to fully show the impact of the proposed backward consistency.**
> > > >
> > > > We agree with the reviewer that if we consider SITCOM without backward consistency and SITCOM where both are run from $t=T$ and with a fixed $K$ for each sampling step, then the run-time will be different. However, the run-time is similar because the reported run-time is the total run-time from $t=T$ for which we run SITCOM to $t=t'+1$. Only at $t=t'$, we do two different optimizations using a fixed $K$. Please note that this the same experimental setting in the second part of Appendix A in our original submission (results of Figure 4).
> > > >
> > > > To fully address the reviewer's concern, we repeat this experiment **with running each optimization until convergence at $t'$**. Empirically, we observed that setting $K$ to 1000 at $t'$ achieves this. Note that The setting here is similar to the setting of the results in Figure 2 of our original submission where more details were provided in the first part of Appendix A.
> > > >
> > > > SR:
> > > > |Method |  $t' = 200$  | $t' = 400$ | $t' = 600$ | $t' = 800$ |
> > > > |---------|------------------|----------------------|--------------|-|
> > > > | SITCOM | 31.32/558.45| 30.78/476.09   |29.89/424.26 | 25.62/256.16 |
> > > > | SITCOM without backward consistency at $t'$ | 25.24/529.56| 24.74/452.22   |17.56/414.45 | 14.45/245.87 |
> > > >
> > > > NDB:
> > > > |Method |  $t' = 200$  | $t' = 400$ | $t' = 600$ | $t' = 800$ |
> > > > |---------|------------------|----------------------|--------------|-|
> > > > | SITCOM | 30.78/544.45| 30.56/483.78   | 28.23/424.69 | 26.78/229.24 |
> > > > | SITCOM without backward consistency at $t'$  | 24.88/529.44| 23.56/464.88   | 21.45/412.68 | 12.25/216.47 |
> > > >
> > > > These results indicate that the proposed backward consistency indeed return improved results when compared to the case where the backward consistency is not used.
> > > >
> > > > ## Consistency of DCDP run-time results.
> > > >
> > > > Thanks for pointing this out. The reviewer is correct that the run-time of DCDP with $N=K=30$ is incorrect. **It is a typo; It should be 24.02 seconds instead of 12.02 seconds**. We have fixed this typo in the above table and, as recommended, carefully double checked the other results. We fixed another typo in the second table of the impact of the backward consistency experiment.
> > > >
> > > > We have also added more DCDP instances to show that the reported results are consistent.

---

> > > > > ### Author Response · Authors · 2024-11-27
> > > > > **Thank you for your response. Part 2/2**
> > > > >
> > > > > ## Benchmarking comparison of the methods under consistent settings and across a unified set of tasks.
> > > > >
> > > > > We agree with the reviewer that a comprehensive benchmarking across a unified set of tasks and settings is preferable. **However, we were selective in the tasks, settings, and datasets for baselines as not all baselines considered 8 image restoration tasks and tested with different noise levels on two datasets**. For example, PR was not considered in DMPlug and ImageNet was not considered in FPS. If we run PR on DMPlug or FPS on ImageNet, we are unsure about the hyper-parameters of other tasks (or datasets) which may not return the best of DMPlug or FPS. **For this reason, we argue that it would be an unfair comparison if we run DMPlug and FPS on PR and ImageNet, respectively**.
> > > > >
> > > > > ## Additional tasks and metrics for Impact of the iterative application of measurement consistency in the first step of SITCOM.
> > > > >
> > > > > We thank the reviewer for their comment. As recommended, we report the results of SR, NDB, PR, and HDR using PSNR, run-time, SSIM, and LPIPS.
> > > > >
> > > > > |$N$ | SR $K=1$ | SR $K=20$ | NDB $K=1$ | NDB $K=20$ |
> > > > > |---------|------|----------|---------|------|
> > > > > |---------|PSNR/SSIM/LPIPS/time|PSNR/SSIM/LPIPS/time|PSNR/SSIM/LPIPS/time|PSNR/SSIM/LPIPS/time|
> > > > > | 10 | 11.28/0.587/0.481/0.99 | 27.75/0.831/0.198/19.12    | 12.12/0.598/0.477/0.99 | 26.92/0.826/0.165/19.02    |
> > > > > | 20 | 11.41/0.589/0.479/2.11 | 30.40/0.866/0.148/35.20    | 12.30/0.602/0.468/2.30 | 30.27/0.862/0.150/36.05    |
> > > > > | 30 | 11.44/0.590/0.477/3.12 | 30.88/0.877/0.142/55.04    |12.32/0.608/0.465/3.40 | 30.72/0.867/0.145/57.05    |
> > > > > | 100 | 17.04/0.651/0.439/9.14 | --    |16.46/0.642/0.447/9.12 | --    |
> > > > > | 1000 | 25.44/0.771/0.261/60.43 | --    |24.89/0.767/0.267/60.45 | --    |
> > > > >
> > > > > |$N$ | PR $K=1$ |  PR $K=20$ | HDR $K=1$ | HDR $K=20$ |
> > > > > |---------|------|----------|---------|------|
> > > > > |---------|PSNR/SSIM/LPIPS/time|PSNR/SSIM/LPIPS/time|PSNR/SSIM/LPIPS/time|PSNR/SSIM/LPIPS/time|
> > > > > | 10 | 10.28/0.547/0.501/1.09 | 28.75/0.864/0.171/19.03    | 11.98/0.594/0.483/1.02 | 26.22/0.812/0.205/19.12    |
> > > > > | 20 | 11.02/0.570/0.491/2.23 | 31.02/0.901/0.132/34.26    | 12.02/0.598/0.471/2.28 | 27.47/0.824/0.178/36.07    |
> > > > > | 30 | 11.34/0.588/0.483/3.25 |31.37/0.907/0.112/54.08    |12.22/0.601/0.468/3.22 | 27.98/0.832/0.162/56.92   |
> > > > > | 100 | 17.07/0.656/0.432/9.08 | --    |15.86/0.622/0.454/9.26 | --    |
> > > > > | 1000 | 23.44/0.751/0.281/60.25 | --    |21.09/0.737/0.287/60.32 | --    |
> > > > >
> > > > > As observed, the LPIPS and SSIM results are consistent with the the reported PSNRs. To this end, we feel that our observations/arguments in this experiment remain valid.
> > > > >
> > > > > ## Presentation of Additional Results. Baselines should have been considered in the original submission.
> > > > >
> > > > > We thank the reviewer for their comment. In our original submission, we feel that we did evaluate our method very well. The reason for this is our consideration of two datasets, two measurement noise levels, 8 tasks (5 linear and 3 non-linear), and 4 baselines that achieve SOTA results (on either PSNR or run-time). See Tables 1 and 2 in DCDP and DAPS.
> > > > >
> > > > > Additionally, only 2 out of the 5 baselines we included in the rebuttal, generally, achieve competitive results on several tasks (DMPlug and PnP-DM). **Note that both of these papers are Neurips24 papers which means that they are very recent**. It was challenging to include these comparisons at the time of the submission. That said, we promise to include these comparisons in a subsequent version of the paper.
> > > > >
> > > > > If the reviewer could elaborate more on which results they find challenging to interpret, we will be glad to address these concerns.
> > > > >
> > > > > ## Analysis of the Proposed Method. The additional results don't fully address my initial concerns.
> > > > >
> > > > > In the original paper, we provided two experiments (with visualizations) about the impact of the backward consistency which is our main contribution/novelty of the paper. In the rebuttal, we provided three additional experiments to show the impact of the other two consistencies in addition to reporting the average results of the backward consistency.
> > > > >
> > > > > If the reviewer still finds any of these challenging to interpret, we will be glad to address any remaining concerns.
> > > > >
> > > > > **We took the opportunity of the extended discussion time to address the reviewer's remaining concerns and include them (along with the previous experiments) in the appendix of the revised paper. We take full responsibility of the previous typo and doubled check all results. Moreover, no changes to main body of the paper. We hope that the reviewer can take another look to see if we addressed their concerns.**

---

> > > > > > ### Author Response · Authors · 2024-12-03
> > > > > > **Thank you for your response**
> > > > > >
> > > > > > ## Additional Experiment for the impact of the backward consistency
> > > > > >
> > > > > > We would like to thank the reviewer for their constructive feedback.
> > > > > >
> > > > > > In our previous reply, we claimed that running SITCOM without the backward consistency from $t=T$ will result in reduced run-times when compared to SITCOM. However, the PSNRs will be lower which shows the impact of the proposed backward consistency. The reduction in run-time, as the reviewer pointed out, is due not needing the backward pass through the DM pre-trained network.
> > > > > >
> > > > > > Our previous comparisons between SITCOM and SITCOM without backward consistency is at $t=t'$ where we run SITCOM from $t=T$ to $t=t'+1$, then perform two separate optimizations. We agree that it might be more informative to just compare the result at convergence (setting $K$ to be sufficiently large to ensure convergence).
> > > > > >
> > > > > > **To fully address the reviewer's concern, here, we report the results of SITCOM vs. SITCOM without backward consistency for $K=1000$ starting from $t=T$ and reporting the results at $t=0$ on two separate optimizations**.
> > > > > >
> > > > > > **SR**: $K = 1000$ at noise level of 0.05
> > > > > > |Method | $N$ | PSNR (dB) | run-time (seconds) |
> > > > > > |---------|------------------|----------------------|--------------|
> > > > > > | SITCOM                 | 20 |  31.34 | 608.25|
> > > > > > | SITCOM without backward consistency  | 20 | 25.97|372.44|
> > > > > > | SITCOM                 | 100 |  31.39 | 2418.25|
> > > > > > | SITCOM without backward consistency  | 100 | 26.10|1405.65|
> > > > > >
> > > > > > As observed, SITCOM without backward consistency indeed requires less run-time but the PSNRs are significantly lower than SITCOM (more than 5 dB for both cases).

---

### Official Review · Reviewer_7KC3 · 2024-11-04

**Soundness:** 2
**Presentation:** 4
**Contribution:** 2
**Rating:** 5
**Confidence:** 4

**Summary:**

The authors identify limitations in current diffusion model-based inverse solvers, noting that the measurement-guidance term often pushes the trajectory toward inconsistency, leading to artifacts in intermediate samples and requiring a large number of correction steps.

To address these issues, they propose three conditions for achieving measurement-consistent diffusion trajectories: (1) standard data manifold measurement consistency, (2) forward diffusion consistency, and (3) backward diffusion consistency.

1. Measurement-consistent: Reconstruction is consistent with the measurements.
2. Forward-consistent: Intermediate samples during the diffusion trajectory should resemble in-distribution samples produced by forward diffusion.
3. Backward-consistent: The measurement-guided output, followed by Tweedie’s denoising, should also be the Tweedie-denoised output of some noisy data.

The authors introduce an algorithm that ensures all three consistencies, called Step-wise Triple-Consistent Sampling (SITCOM). The proposed method outperforms existing approaches in various image inverse problems.

**Strengths:**

The paper is well-written and easy to follow.
- It clearly motivates the issue of inconsistency in current models and provides an intuitive solution to address these inconsistencies.
- The proposed algorithm, which integrates these solutions, demonstrates improved performance over existing methods.

The authors showcase the applicability of their method across various image inverse problems.

**Weaknesses:**

The claims about consistency are not fully substantiated. It remains unclear if the proposed method genuinely addresses all three inconsistencies.
- Experimental validation is needed to quantify how effectively the method mitigates these inconsistencies. For example, could experiments be designed to measure the degree to which each inconsistency is reduced?

The impact of each individual component in the method is not verified.
- How would the results change if one component were removed?
- Additionally, could similar components be added to other algorithms, such as DPS or DDNM, to reduce inconsistencies in those methods as well? Would this approach similarly alleviate inconsistencies?

**Questions:**

The questions are integrated in the section above.

---

> ### Author Response · Authors · 2024-11-21
> **Reviewer 7KC3**
>
> We thank the reviewer for their comments.
>
> **[C1/C2] The impact of each individual component in SITCOM.**
>
> Thank you for your suggestion. Please see the first part of the global response.
>
> **[C3] The possibility of adding SITCOM components to DPS or DDNM.**
>
> We would like to thank the reviewer for their question. In the first study of our global response, we show the impact of the iterative application of measurement consistency. In this experiment, we compared SITCOM (i.e., $K>1$) with $K=1$ which is DPS plus the resampling formula (for forward consistency). Here, $K$ is the number of gradient updates per sampling step.
>
> The results indicate that DPS, as is already established, requires a relatively very large $N$ to obtain descent results. We note that including the resampling in DPS's formula violate their theoretical results.
>
> For the case of DDNM, which is only applicable to linear IPs, the algorithm initially estimate an image at every reverse step using Tweedie's formula without measurement consistency. Subsequently, DDNM enforces data consistency by refining this estimate using the pseudoinverse of the forward operator. In order to remap the image back to $t-1$, the authors use the standard sampling (Eq.(14) of their paper and Eq.(7) in our paper) instead of the resampling formula for partially applying the forward consistency (Eq.(8) or Eq.(S3) of our paper).
>
> While we conjecture that using the resampling for remapping would improve the performance in DDNM, we don't believe that DDNM can apply our proposed backward consistency (with the step-wise network regularization) as they apply measurement consistency differently.

---

> > ### Author Response · Authors · 2024-11-25
> > **A friendly and gentle reminder**
> >
> > We would like to express our sincere gratitude to the reviewer once again for their insightful comments.
> >
> > As the open discussion period is drawing to a close, we would be deeply grateful if the reviewer could kindly respond to our rebuttal. This would provide us with the opportunity to address or clarify any remaining concerns thoroughly.

---

> ### Comment · Reviewer_7KC3 · 2024-12-02
>
> Thank you for addressing the reviewers’ concerns. While the authors have successfully addressed my specific concern about the work, the discussion with another reviewer remains unresolved. Therefore, I maintain my previous score.

---

> > ### Author Response · Authors · 2024-12-03
> > **Thank you for your response**
> >
> > We would like to thank the reviewer for their thoughtful response. We are pleased to hear that our revisions have successfully addressed their specific concerns.
> >
> > Regarding the "unresolved" discussion, we kindly note that, in our effort to thoroughly address the comments from the other reviewer, we have provided two responses to their remaining concerns over the past few days.

---

### Official Review · Reviewer_uETD · 2024-11-04

**Soundness:** 2
**Presentation:** 3
**Contribution:** 2
**Rating:** 3
**Confidence:** 4

**Summary:**

This paper presents an optimization-based algorithm, SITCOM, to solve inverse problems with a pretrained diffusion model. The authors state three conditions to hold at each sampling step of the diffusion model. SITCOM optimizes measurement consistency on $x_t$ and uses a resampling procedure to enforce the consistency conditions. Experiments show that SITCOM performs better than existing methods in several image restoration tasks.

**Strengths:**

- The paper is well-presented, with a clear and structured approach.
- The proposed algorithm introduces "forward consistency" and "backward consistency," which is a novel framework that may interest a broader research community.
- Experimental results on image restoration tasks are promising.

**Weaknesses:**

- The paper does not provide rigorous guarantees for the output of SITCOM, whereas several prior methods offer assurances on correct sampling from the desired posterior distribution [2,3,4,7].
- The technical novelty appears limited. The optimization of $x_t$ for measurement consistency using Tweedie’s approximation directly resembles DPS, while the resampling technique is derived from ReSample. The resulting algorithm appears to be a straightforward combination of existing methods.
- This paper lacks a comparison to many relevant works [1~8]. Some of them provide asymptotic exact posterior sampling [2,3,4,7], while [1] appears to achieve the proposed "backward consistency" as well. It seems strange that SITCOM is only compared with DPS, DDNM, and two closely related algorithms (DAPS, DCDP), which seemingly have not undergone peer review.
- The stopping criterion for preventing noise overfitting needs further justification. Given that the actual noise level $\sigma_y$ is typically unknown in practice, an ablation study on the algorithm's sensitivity to $\delta$ would be valuable.

### Reference
[1] Wang et al. "A Plug-in Method for Solving Inverse Problems with Diffusion Model." In NeurIPS 2024.

[2] Dou et al. "Diffusion Posterior Sampling for Linear Inverse Problem Solving: A Filtering Perspective." ICLR 2023.

[3] Gabriel et al. "Monte Carlo guided Denoising Diffusion models for Bayesian linear inverse problems." ICLR 2023.

[4] Wu et al. "Principled Probabilistic Imaging using Diffusion Models as Plug-and-Play Priors." In NeurIPS 2024.

[5] Rout et al. "Solving linear inverse problems provably via posterior sampling with latent diffusion models." In NeurIPS 2023.

[6] Song et al. "Solving Inverse Problems with Latent Diffusion Models via Hard Data Consistency." In ICLR 2024.

[7] Wu et al. "Practical and Asymptotically Exact Conditional Sampling in Diffusion Models." In NeurIPS 2023.

[8] Rout et al. "Beyond first-order Tweedie: Solving inverse problems using latent diffusion." In CVPR 2024.

**Questions:**

- How is the stopping criterion determined in practice? Does different $\delta$ influence the final output of SITCOM?

---

> ### Author Response · Authors · 2024-11-21
> **Reviewer uETD: Part 1/2**
>
> We thank the reviewer for their comments.
>
> **[C1] Guarantees for the output of SITCOM. Assurances on correct sampling from the posterior distribution like in [2,3,4,7]**.
>
> We agree that having a theoretical guarantee is indeed preferable. Previous studies [2,3,4,7] have provided such guarantees for sampling from the posterior distribution, but their performance is not optimal. In contrast, other methods, such as ReSample, DAPS, and DCDP, which leverage step-wise data consistency, achieve better results. However, establishing a performance guarantee for these methods is challenging. There seems to be some trade-off. Our approach falls into the latter category **with the major distinction of incorporating the step-wise backward consistency** through the proposed network regularization. To address this limitation, we conducted extensive numerical experiments and ablation studies to evaluate our method. Please see our global response.
>
> **[C2] Resemblance to DPS and ReSample and scope of technical novelty. [Continued]**
>
> We thank the reviewer for their comment. Many DM-based IP samplers/solvers share similarities, as summarized in the following table. Previous works do indeed contain data and forward consistencies. However, we argue that these alone are insufficient to achieve an improved performance. *We believe that incorporating backward consistency can further enhance performance (as supported quantitatively and computationally)*.
>
> |Method | Step-wise Measurement Consistency | Step-wise network regularization (backward consistency in Eqs.(S1) and (S2))| Forward Consistency (Eq.(S1) regularization and initialization + Eq. (S3)) |
> |---------|------|----------|-|
> | SITCOM (Ours)   | $\checkmark$    |  $\checkmark$ | $\checkmark$ |
> |DPS  |  $\times$    |  $\times$ |  $\times$
> | ReSample  | $\checkmark$    | $\times$ |  $\checkmark$ |
> | DAPS  | $\checkmark$    | $\times$ |  $\checkmark$ |
> | DCDP  | $\checkmark$    | $\times$ |  $\checkmark$ |
> | DMPlug   | $\checkmark$    | $\times$ |  $\checkmark$ |
> | PnP-DM   | $\checkmark$   | $\times$ |  $\checkmark$|
>
> The technical novelty of SITCOM can be highlighted as:
> - **Formulating the three consistencies**: We present a systematic formulation of the conditions where the goal is to improve the accuracy of each intermediate reconstruction, thereby reducing the number of sampling steps required.
> - **Introducing the step-wise Backward consistency**: We introduce step-wise network regularization for the backward consistency by leveraging the implicit bias of the network at each iteration. In this rebuttal, we present additional numerical results demonstrating the benefits of enforcing the proposed backward consistency (see the third experiment of the second part of the global response).
> - **Integrating all consistencies:** The formulation, initialization, and regularization of the optimization problem in Eq.(S1) along with the preceding two steps ensure that all consistencies are enforced without violation. Note that the initialization (and regularization) in Eq.(S1) is partially used to ensure closeness such that the forward consistency is satisfied.
>
> It is worth noting that while some previous works, such as RED-diff and DMPlug, also utilize the implicit bias of the network, *they adopt the full diffusion process as a regularizer, applied only once*. In contrast, our method uses the neural network as the regularizer at each iteration and focuses specifically on reducing the number of sampling steps for a given level of accuracy. We compared our approach with these methods to highlight this advantage (see the first part of the global response). Please refer to the highlighted changes on pages 5 and 6 of the revised manuscript.

---

> ### Author Response · Authors · 2024-11-21
> **Reviewer uETD: Part 2/2**
>
> **[C3] (i) Comparison Results. (ii) Asymptotic exact posterior sampling in [2,3,4,7]. (iii) Backward consistency in DPS. (iv) Choosing DCDP and DAPS as baselines.**
>
> We thank the reviewer for this comment.
>
> (i) Please see our global response. For the suggested latent solvers, please see our response to Comment 3 of Reviewer iQjf.
>
> (ii) We agree with the reviewer that [2,3,7] provide asymptotic exact posterior sampling. However, this exact posterior sampling is achieved based on assumptions that may not be fully realistic/possible, and consider other sampling techniques to estimate the intractable term. For example, FPS [2] is built based on the assumption that the pre-trained score model is a *perfect* estimator of the log likelihood (see Assumption 4.1 in [2]). In MCGdiff [3], the asymptotic posterior is obtained through Sequential Monte Carlo (SMC) sampling which are also estimators used when the posterior is infeasible due to complexity. Furthermore, the methods in [2,3] are only applicable to linear imaging IPs. Similar to MCGdiff, the authors in [7] introduced Twisted Diffusion Sampling (TDS) which is also based on MCS sampling. TDS is a general conditional diffusion sampler where the authors considered two problems: Image in-painting on MNIST (not a high dimensional dataset), and motif-scaffolding (a task in protein design) which is not an imaging inverse problem. In PnP-DM [4], the authors provide insights on the non-asymptotic behavior of the proposed algorithm for the which the method uses Split Gibbs Sampling. Therefore, we believe that trying to approximate exact posterior sampling under some assumption and/or using other estimators may not be sufficient or the best way to obtain improved reconstructions as supported by the results.
>
> (iii) Please see our response to the previous comment for the backward consistency in DPS.
>
> (iv) We agree with the reviewer that two key baselines in our paper, DCDP and DAPS, are unpublished yet. However, since we ran their codes and verified their results—and given the rapid pace of this field—we believe they remain valid baselines.
>
> Generally, to select baselines, we focused on methods addressing various linear and non-linear IPs, using diverse datasets and pre-trained models, accounting for measurement noise, and demonstrating superior reconstruction and/or run-time performance.
>
> Notably, DAPS achieves SOTA PSNR results, while DCDP offers competitive PSNRs with reduced run-time. Additionally, we compare to DMPlug (NeurIPS24) in the global response which also achieved very competitive results.
>
> **[C4/Q1] Sensitivity to choice of $\delta$ and how it is determined in practice.**
>
> Thank you for comment/Question. In SITCOM, we proposed to use a stopping criterion to prevent the noise overfitting, determined by $\sigma_y$. In Appendix F.4, we show the impact of using the stopping criterion vs. not using it. In this experiment, as suggested by the reviewers, we run SITCOM using different values of $\delta$ and report the average PSNR values and run-time (in seconds).
>
> We use $N=K=20$ and $\sigma_y = 0.05$ and set $\delta$ to values above and below $\sigma_y$. Results for BIP (linear IP) and NDB(non-linear IP) are given in the following table where we use 20 FFHQ test images with $m=256\times 256 \times 3$. The first row shows the values of $a$ for which $\delta = a\sqrt{m}$.
>
> |Task |  $0.01$ | $0.025$ |$0.04$ |$0.05$ |$0.051$ |$0.055$ |$0.1$ |$0.5$ |
> |---------|------|----------|-|-|-|-|-|-|
> | Super-Resolution    | 27.80/40.20    | 29.02/38.41 |30.22/36.26    | 30.39/28.52 |30.40/28.26    | 30.37/24.10 |28.87/22.12    | 23.10/15.87 |
> | Non-uniform  Deblurring  | 28.40/40.40    | 29.80/37.32 | 29.82/35.12    | 30.12/32.12 |30.21/29.46    | 30.20/27.44 |29.12/22.05    | 25.79/15.10 |
>
> As observed, the PSNR values vary between 29.02 to 30.37 (resp. 29.80 to 30.20) only for BIP (resp. NDB) with stopping criterion between 0.025 to 0.055 which **indicates that even if values lower (or slightly higher) than the measurement noise level are selected, SITCOM can still perform reasonably well**.
>
> This means that if we use classical methos to approximate/estimate the noise, we can achieve good results. In practice, we can use classical methods (such as Liu et al., 2006 and Chen et al., 2015) to estimate the noise present in the measurements which we discuss in the Limitations and Future Work (Appendix B).
>
> We note that other works, such as DAPS and PGDM [B], also use $\sigma_{y}$ but not to estimate the stopping criterion. In these papers, $\sigma_{y}$ is encoded in the updates of their algorithms (see Eq. (7) in PGDM and Eq. (9) in DAPS).
>
> [B] Pseudoinverse-guided diffusion models for inverse problems. ICLR, 2023.

---

> > ### Author Response · Authors · 2024-11-25
> > **A friendly and gentle reminder**
> >
> > We would like to express our sincere gratitude to the reviewer once again for their insightful comments.
> >
> > As the open discussion period is drawing to a close, we would be deeply grateful if the reviewer could kindly respond to our rebuttal. This would provide us with the opportunity to address or clarify any remaining concerns thoroughly.

---

### Official Review · Reviewer_4Vbo · 2024-11-05

**Soundness:** 2
**Presentation:** 2
**Contribution:** 2
**Rating:** 5
**Confidence:** 4

**Summary:**

This paper addresses inverse problems using a plug-and-play approach with diffusion models, focusing on reducing the low sampling speeds that are characteristic of iterative diffusion models in these settings. To this end, the authors propose a "triple consistency" framework, adding a backward consistency component to the already established data and forward consistencies. The backward consistency aims to ensure that the solution obtained after data-consistency remains a valid diffusion model solution, essentially acting as a projection onto the intersection of data-consistent and diffusion model-consistent solutions.

The paper provides experimental results on both linear and nonlinear tasks and compares the proposed method with other existing techniques, showing performance improvements for certain tasks.

**Strengths:**

- The paper addresses a timely and important problem in inverse problem-solving using diffusion models.

- Experiments cover a variety of linear and nonlinear tasks, giving a broad perspective on the method's applicability.

**Weaknesses:**

Modeling of Backward Consistency: The concept of backward consistency is interesting, but its implementation feels somewhat ad hoc. While it resembles a projection onto the intersection of data and prior-consistent solutions, this usually requires multiple iterations to reach a meaningful intersection. Here, however, only a single iteration is employed, which may be insufficient.

Alternative Approach: The backward consistency term could potentially be better represented as a fixed-point condition for the denoiser. This approach might be more systematic, aligning with methods like RED-diff, which regularize through denoising consistency.
Sufficiency of Eq. (4): Additionally, it's unclear if enforcing Eq. (4) (the Tweetie consistency) is sufficient to ensure backward consistency. This claim requires more justification.

Experimental Limitations:
- Comparison Scope: Tables 1 and 2 lack comparisons with key existing methods. Notably, the absence of comparisons with RED-diff and PGDM in Table 2 weakens the empirical analysis.

- Unfair Comparison: Simply reusing the hyperparameters from existing methods without adjusting them for the current dataset results in an unfair comparison. This could particularly impact performance for methods like DPS, RED-diff, and PGDM, which may not perform optimally without parameter tuning.

- Sampling Efficiency: A central claim of this work is the reduced number of iterations required by the proposed method. However, the paper lacks concrete experimental evidence to substantiate this, such as timing results that would demonstrate faster sampling speeds in seconds. Given the emphasis on efficiency, readers would expect clear evidence supporting this claim.

[PGDM] Song, J., Vahdat, A., Mardani, M., & Kautz, J. (2023, May). Pseudoinverse-guided diffusion models for inverse problems. In International Conference on Learning Representations.

**Questions:**

- How sensitive is the performance with respect to the parameter λ?

see the weakness part for more comments

---

> ### Author Response · Authors · 2024-11-21
> **Reviewer 4Vbo**
>
> We would like to thank the reviewer for their comments.
>
> **[C1] Modeling of Backward Consistency & the projection onto an intersection of data- and prior-consistent solutions.**
>
> The reviewer is right that if we want to find a point in the intersection of two sets $C_1\cap C_2$, a common approach is using alternating projection. However, due to the specific structure of the backward consistent set, we can avoid doing the alternating projection and instead solve a single optimization problem that includes the violation of both constraints as penalty terms. Essentially, we are minimizing a smoother version of the objective $1_{C_1}(x)+1_{C_2}(x)$, where $1_{C}(\cdot)$ is the indicator function that equals $\infty$ when $x$ is outside $C$ and 0 when $x$ is inside $C$. More specifically, $x$ is defined to be **backward-consistent** if it can be expressed as $x= f_{\theta}(z)$ with some $z$. It is data-consistent if $\mathcal{A}(x) =y$ (assuming no noise). Combining these, we then obtain the following direct optimization problem:
> $\min_{x,z} \\{ \||\mathcal{A}(x)-y\||^2 \text{     s.t.    } x= f_\theta(z) \\}$,
> solving which enforces both consistencies. Since $x$ is a function of $z$, we can eliminate it and arrive at $\min_{z} \{ \||\mathcal{A}(f_\theta(z))-y\||^2\}$, which is the optimization problem in our proposed algorithm without the regularization term.
>
> **[C2] The denoiser fixed-point condition for backward consistency, and the clarity of the Tweedie's consistency.**
>
> The triple consistency discussed in the paper is **step-wise**, meaning that they are enforced at each sampling step. For instance, consider data consistency: DPS enforces global data consistency, i.e., the $\hat{x}_0$ is only consistent with the data at the last step when $t=0$, while ReSample enforces it in many steps (indices in set $C$ of their algorithm), from $t=T$ down to $t=0$. This stepwise data consistency enforcement is one of the main factors driving the improved results of ReSample. Similarly, our backward consistency (as defined in Definition 1) is also step-wise with the **major distinction that we require the reconstruction $\hat{x}_0$ at each step to be consistent with the network regularization and Tweedie's formula**.
>
> This contrasts with RED-diff and PGDM, which enforce *global* network regularization which uses the whole diffusion process as a regularizer. The global regularization approach, however, inevitably is not as effective for the improvement of the reconstruction quality (see the additional comparison results that we include in the global response).
>
> Indeed, while Tweedie's formula is one way to incorporate step-wise network regularization, it may not be the only (or the best) approach in terms of restoration quality, but it is indeed very efficient in terms of reducing the run-time while maintaining competitive PSNRs. We plan to explore alternative methods in future work.
>
> **[C3] Additional Comparison Results.**
> Please see the global response.
>
> **[C4] Hyper-parameters of existing baselines.**
> For the comparisons in our paper (and rebuttal), we would like to point out that we have utilized the same datasets, pre-trained DMs, and the recommended (fine-tuned) hyperparameters as specified in the baseline papers. Furthermore, we downloaded and executed their code, verifying that the recommended hyperparameters align with the results reported in their respective papers.
>
> **[C5] Sampling efficiency and Run-time results.**
> We agree with the reviewer that the main motivation of introducing SITCOM is to obtain an efficient sampler for solving IPs using DMs while maintaining competitive or leading PSNRs. In our paper, we report the runtime (average and standard deviation in minutes) in the 6th and 10th columns of Tables 1 and 2 in the original submission and Tables 1 and 4 in the revised paper.
>
> **[Q1] SITCOM's sensitivity to the choice of $\lambda$.**
> In Table 5 of Appendix F.2 (which is now Table 7 in the revised paper), we report PSNR results of SITCOM with different values of $\lambda$ using 3 linear tasks and one non-linear task. Here, we include PSNR and run-time results of Gaussian Deblurring. The remaining tasks are in (https://anonymous.4open.science/r/SITCOM-7539/Extended_Ablation.md).
>
> | $\lambda$ | PSNR    | Runtime (sec) |
> |-----------|---------|---------------|
> | $0.05$    | 28.846  | 53.16         |
> | $0.5$     | 29.033  | 53.64         |
> | $1.0$     | 29.231  | 58.26         |
> | $1.5$     | 29.230  | 60.48         |
>
> As observed, the PSNR values do not significantly change, indicating that *SITCOM is not sensitive* to the choice of $\lambda$. We conjecture that initializing the optimization variable in Eq.(S1) as $x_t$ (Step 2 of our Algorithm) plays a bigger role in ensuring the closeness stated in Condition 3.

---

> > ### Author Response · Authors · 2024-11-25
> > **A friendly and gentle reminder**
> >
> > We would like to express our sincere gratitude to the reviewer once again for their insightful comments.
> >
> > As the open discussion period is drawing to a close, we would be deeply grateful if the reviewer could kindly respond to our rebuttal. This would provide us with the opportunity to address or clarify any remaining concerns thoroughly.

---

### Author Response · Authors · 2024-11-21
**Global Response: Part 1/2**

We would like to thank the reviewers for their constructive feedback. The global response includes comparisons with suggested baselines and studies on the impact of each consistency in SITCOM, as recommended.

## Comparison Results:

Here, we note that:

- We either run the codes of the suggested baselines or use the results reported by the authors. Specifically, we run the code of PGDM [R.iQjf;2], DMPlug [R.uETD;1], and RED-diff [R.4Vbo]. We copy the results of FPS [R.uETD;2] and PnP-DM [R.uETD;4] from their papers. Due to the time limit of the rebuttal, we promise to run and include the results of the remaining baselines in a subsequent version of our paper.
- Tasks are chosen based on the baselines’ evaluations, assuming fine-tuned parameters in the authors' codes. For instance, we omit phase retrieval for DMPlug as it was not evaluated in their work.
- For baselines not run, we report results using the same pre-trained model, test images, task settings, noise levels, and metrics, noting that some methods use only LPIPS, not PSNR or SSIM.
- We prioritized running baselines with leading results to compare their runtime and PSNRs with SITCOM.
- We use SR, BIP, RIP, GDB, MDB, PR, NDB, and HDR to denote super resolution, box-inpainting, random inpainting, Gaussian deblurring, motion deblurring, phase retreival, non-linear deblurring, and high dynamic range.

### DMPlug (NeurIPS24):

- **Dataset**: FFHQ. **Metric**: PSNR/run-time (seconds). **Noise level**: 0.05.

|Method |  SR | RIP | GDB | NDB |
|---------|------|----------|-|-|
| DMPlug     | 30.18/68.20    |  31.02/72.02 | 29.79/65.44 | **30.31**/182.40 |
| SITCOM (Ours)   | **30.68/32.25**    |  **32.05/30.02** | **30.25/36.27** | 30.12/**35.43** |

As observed, we achieve better PSNR results for all tasks other than NDB (where we underperform by 0.19dB). When compared to DMPlug we require nearly half the run-time for linear IPs and approximately 20% of the time for NDB.

### RED-diff (ICLR24):

- **Dataset**: ImageNet. **Metric**: PSNR/run-time (seconds). **Noise level**: 0.05

|Method |  SR | MDB | HDR |
|---------|------|----------|-|
| RED-diff     | 24.89/18.02    |  27.35/19.02 | 23.45/24.40 |
| SITCOM (Default)   | 26.35/62.02    |  28.65/62.04 | 26.97/62.5 |
| SITCOM $(N,K)=(8,8)$ | 24.99/17.80	|  27.38/17.92 | 25.24/20.12 |

As observed, given the same PSNR, for SR and MDB, SITCOM requires less run-time, For HDR, we achieve higher PSNR while requiring less run-time. Furthermore, in SITCOM, an external denoiser is not needed.

### PGDM (ICLR23):

- **Dataset**: ImageNet. **Metric**: PSNR/run-time (seconds). **Noise level**: 0.05

|Method |  SR | MDB |
|---------|------|----------|
| PGDM     | 25.22/26.20    |  27.48/24.10 |
| SITCOM (Default)   | 26.35/62.02    |  28.65/62.04 |
| SITCOM $(N,K)=(10,10)$   | 25.15/19.72	|  27.55/19.65 |

As observed, given the same PSNR, SITCOM requires less run-time. Also, our default $N$ and $K$ achieves more than 1dB improvement.

### FPS (ICLR23):

- **Dataset**: FFHQ. **Metric**: LPIPS. **Noise level**: 0.05. **Source**: Table 1 and Table 2 of FPS.

|Method |  SR | BIP | GDB | RIP | MDB |
|---------|------|----------|-|-|-|
| FPS     | 0.212    |  0.141 | 0.248 | 0.265 | 0.221 |
| FPS-SMC     | 0.21    |  0.15 | 0.253 | 0.275 | 0.227 |
| SITCOM (Ours)  | **0.142**    |  **0.121** | **0.135** | **0.095** | **0.148** |

- **Dataset**: ImageNet. **Metric**: LPIPS. **Source**: Table 1 and Table 2 of FPS.

|Method |  SR | BIP | GDB | RIP | MDB |
|---------|------|----------|-|-|-|
| FPS     | 0.329   |  **0.204** | 0.396 | 0.325 | 0.37 |
| FPS-SMC     | 0.316    | 0.212 | 0.403 | 0.328 | 0.365 |
| SITCOM (Ours)  | **0.232**    |  0.214 | **0.236** | **0.127** | **0.189** |

As observed, SITCOM outperforms FPS in all tasks other than BIP for the ImageNet dataset which we under-perform by 0.01 (in LPIPS score).

### PnP-DM (NeurIPS24):

- **Dataset**: FFHQ. **Metric**: PSNR. **Noise level**: 0.01. **Source**: Table 1 and Table 2 of PnP-DM.

|Method |  GDB | MDB | SR | PR |
|---------|------|----------|-|-|
| PnP-DM (VP)     | 29.46    |  30.06 | 29.4 | 30.36 |
| PnP-DM (VE)     | 29.65    | 30.38 | 29.57 | 29.88 |
| PnP-DM (iDDPM)  | 29.6     | 30.26 | 29.53 | 30.61 |
| PnP-DM (EDM)    | 29.66    | 30.35 | 29.6 | 31.14 |
| SITCOM (Ours)   | **32.12**    |  **32.34** | **30.95** | **31.88** |

As observed, SITCOM outperforms PnP-DM in all tasks considered.

---

> ### Author Response · Authors · 2024-11-21
> **Global Response: Part 2/2**
>
> ## 1: Impact of the iterative application of measurement consistency in the first step of SITCOM:
> This experiment examines the impact of multiple gradient steps for measurement consistency (Condition 1). We run SITCOM with $K=1$ vs. $K>1$ using different sampling steps ($N$) for two tasks (linear and non-linear), fixing steps 2 and 3 while varying step 1. The study uses 20 FFHQ images with $\sigma_{y} = 0.05$.
>
> For $K=1$, only one gradient update is applied for data consistency (as in DPS), while $K=20$ (converged) strictly enforces the data consistency term.
>
> Average PSNR and run-time (in seconds) are given in the following table.
>
> |$N$ | SR $K=1$ | SR $K=20$ | NDB $K=1$ | NDB $K=20$ |
> |---------|------|----------|---------|------|
> | 10 | 11.28/0.99 | 27.75/19.12    | 12.12/0.99 | 26.92/19.02    |
> | 20 | 11.41/2.11 | 30.40/35.20    | 12.30/2.30 | 30.27/36.05    |
> | 30 | 11.44/3.12 | 30.88/55.04    |12.32/3.40 | 30.72/57.05    |
> | 100 | 17.04/9.14 | --    |16.46/9,12 | --    |
> | 1000 | 25.44/60.43 | --    |24.89/60.45 | --    |
>
> The case with $K=1$ can be thought of as DPS plus the resampling formula.
>
> Using multiple gradient steps yields the best results. For $K=1$, achieving decent results requires $N=1000$ with a learning rate of 10, leading to longer runtimes. Extended ablation studies for $N$ and $K$ are included in our response to [R. qJJQ;Q3].
>
> ## 2: Impact of the adopted resampling formula for the Forward Consistency in SITCOM (i.e., Step 3):
> This experiment examines the impact of the resampling formula for enforcing forward consistency in step 3 of SITCOM. Specifically, we compare SITCOM using Eq.(S3) vs. Eq.(7) for remapping to time $t-1$ in step 3. Using Eq.(7) corresponds to standard DM sampling but replaces $\hat{x}_0$ with $\hat{x}'_0$ (from Eq.(S2)).
>
> We note that the forward consistency in SITCOM is enforced via the regularization term in step 1 (with the initialization of Eq.(S1)) and resampling in Eq.(S3), similar to ReSample.
>
> We use 20 FFHQ test images with $\sigma_{y} = 0.05$, setting $N=K=20$. The table below shows average PSNR/runtime (in seconds) for one linear and one non-linear task.
>
> |Setting |  Mapping to $t-1$  | SR | NDB |
> |---------|------------------|----------------------|--------------|
> | SITCOM | Eq.(S3)| 30.40/35.20    | 30.27/36.05 |
> | SITCOM **without** resampling | Eq.(7) | 20.78/35.12    | 22.42/36.22 |
>
>
> As observed, utilizing Eq.(S3) and the regularization/initialization in Eq.(S1), indicating that this is the best approach for the forward consistency.
>
> ## 3: Impact of the Backward Consistency:
>
> In SITCOM, we apply a **step-wise network regularization** for the backward consistency such that we fully exploit the implicit regularization of the network. Removing the step-wise network regularization is equivalent to removing the requirement for $\hat{x}_0 = f(v_t;\theta,t)$, which makes $x_0$ a free variable. This reduces Eq.(S1) to the optimization in Eq.(5).
>
> To this end, we compare optimizing over the input of the DM (as in SITCOM) with optimizing over the output of the DM (i.e., Eq.(5)) at sampling steps $t' \in \\{200, 400, 600, 800\\}$. Specifically, we run SITCOM from $t=T$ to $t=t'+1$. Then, at $t=t'$, we perform two separate optimizations with initializing the optimization variable as $x_{t'}$ : One over the input and another over the output using the same number of optimization steps, $K=20$, with the same noise setting as above.
>
> SR:
> |Method |  $t' = 200$  | $t' = 400$ | $t' = 600$ | $t' = 800$ |
> |---------|------------------|----------------------|--------------|-|
> | SITCOM | 30.22/30.45| 28.78/26.09   |25.45/14.26 | 22.22/10.16 |
> | SITCOM without Backward Consistency  | 15.24/29.56| 13.24/25.12   |11.56/14.15 | 9.45/7.87 |
>
> NDB:
> |Method |  $t' = 200$  | $t' = 400$ | $t' = 600$ | $t' = 800$ |
> |---------|------------------|----------------------|--------------|-|
> | SITCOM | 29.78/29.45| 27.56/23.78   | 24.23/13.69 | 21.78/10.16 |
> |SITCOM without Backward Consistency  | 14.79/29.04| 13.56/23.48   | 11.45/12.88 | 8.25/9.47 |
>
> As observed, given the same run-time (determined by the number of gradient updates), optimizing over the input consistently achieves better results, highlighting the impact of the proposed step-wise network regularization for the backward consistency. We note that the lower PSNRs is due not running the algorithm until convergence.
>
> It is worth noting that while some previous works, such as RED-diff and DMPlug, also utilize the implicit bias of the network, they adopt the full diffusion process as a regularizer, applied only once. In contrast, our method uses the neural network as the regularizer at each iteration and focuses specifically on reducing the number of sampling steps for a given level of accuracy.

---

### Meta-Review · Area_Chair_KiN1 · 2024-12-20

**Metareview:**

Summary. This paper proposes to solve inverse problems using a plug-and-play approach with diffusion models. The proposed "triple consistency" framework adds a backward consistency component with already established data and forward consistencies. The backward consistency aims to ensure that the solution obtained after data-consistency remains a valid diffusion model solution.

Strengths. The paper addresses a timely and important problem in inverse problem-solving using diffusion models. Experiments cover a variety of linear and nonlinear tasks, giving a broad perspective on the method's applicability. The paper is well-written.

Weaknesses. Technical novelty of the paper appears limited. The proposed backward consistency approach seems ad hoc. The paper lacks comparison with some of the existing methods and additional experiments discussed during rebuttal show mixed results. Some reviewers argued that some of the comparisons seem unfair or unclear. The claims about consistency are not fully substantiated. It remains unclear if the proposed method genuinely addresses all three inconsistencies.

Missing.
The paper shares some similarities with other methods (mainly DCDP). A clear description of the differences and their importance is missing.
The paper is mainly missing some convincing comparisons with existing methods that resemble the proposed method.
The claims about consistency are not fully substantiated. It remains unclear if the proposed method genuinely addresses all three inconsistencies.

Justification.
Missing and mixed comparisons with related methods and discussion about similarities with DCDP were most important reasons for my decision.

**Additional Comments On Reviewer Discussion:**

The paper had an extensive discussion among authors and reviewers.

The reviewers mainly raised concerns about comparisons with existing methods, ablation studies, and significance/justification of the proposed method.

Authors provided detailed responses with additional results. The results raised further questions about fairness of the comparisons.
The reviewers maintained their scores (leaning reject).

---

### Decision · Program_Chairs · 2025-01-22

Reject